# Isoform-resolved correlation analysis between mRNA abundance regulation and protein level degradation

Barbora Salovska[1,2] (ID), Hongwen Zhu[3], Tejas Gandhi[4] (ID), Max Frank[5], Wenxue Li[1],
George Rosenberger[6] (ID), Chongde Wu[1], Pierre-Luc Germain[7,8] (ID), Hu Zhou[3] (ID), Zdenek Hodny[2],
Lukas Reiter[4] (ID) & Yansheng Liu[1,9,*] (ID)

## Abstract

Profiling of biological relationships between different molecular layers dissects regulatory mechanisms that ultimately determine cellular function. To thoroughly assess the role of protein post-translational turnover, we devised a strategy combining pulse stable isotope-labeled amino acids in cells (pSILAC), data-independent acquisition mass spectrometry (DIA-MS), and a novel data analysis framework that resolves protein degradation rate on the level of mRNA alternative splicing isoforms and isoform groups. We demonstrated our approach by the genome-wide correlation analysis between mRNA amounts and protein degradation across different strains of HeLa cells that harbor a high grade of gene dosage variation. The dataset revealed that specific biological processes, cellular organelles, spatial compartments of organelles, and individual protein isoforms of the same genes could have distinctive degradation rate. The protein degradation diversity thus dissects the corresponding buffering or concerting protein turnover control across cancer cell lines. The data further indicate that specific mRNA splicing events such as intron retention significantly impact the protein abundance levels. Our findings support the tight association between transcriptome variability and proteostasis and provide a methodological foundation for studying functional protein degradation.

**Keywords** alternative splicing; DIA mass spectrometry; protein turnover; proteomics; pulsed SILAC
**Subject Categories** Post-translational Modifications & Proteolysis; Translation & Protein Quality; Proteomics
**Mol Syst Biol. (2020) 16: e9170**

## Introduction

Correlation analysis (Altman & Krzywinski, 2015) has been widely used in biological and biomedical reports due to the easily interpretable message it delivers, i.e., whether there is a positive or negative association between the two biological traits and whether this association is statistically significant. As a prominent example, the mRNA–protein correlation has been frequently investigated to reveal to what extent mRNA levels can predict protein levels (Vogel & Marcotte, 2012; Jovanovic et al, 2015; Edfors et al, 2016; Liu et al, 2016; Silva & Vogel, 2016; Fortelny et al, 2017; Franks et al, 2017).

To thoroughly investigate the mRNA–protein correlation, it is indispensable to distinguish between two scales: the absolute and the relative (Liu et al, 2016). While the *absolute correlation* is performed across genes in one particular sample, the *relative correlation* must be performed across samples, e.g., as a correlation of fold changes between conditions. Furthermore, the *relative* correlation analysis can be performed in a *gene-specific* manner (Fortelny et al, 2017) to understand gene-specific properties. The *absolute* correlation is known to be heavily driven by the genome-wide variation of mRNA and protein abundances (i.e., copies per cell). Thus, absolute mRNA–protein correlation is usually high if both transcriptomic and proteomic measurements are precise. On the other hand, the *relative* correlation of the fold changes per gene may directly reflect the significance of the post-transcriptional regulation between different conditions (Liu et al, 2016; Silva & Vogel, 2016; Fortelny et al, 2017).

With the technology development, many gene-specific traits, other than mRNA and protein quantities, can be measured at the genome level, or in large scale. For example, the protein-specific degradation rates ($k_{loss}$ as a proxy in steady state cells in this report, Materials and Methods) can now be quantified by "pulse labeling"

1   Yale Cancer Biology Institute, Yale University, West Haven, CT, USA
2   Department of Genome Integrity, Institute of Molecular Genetics of the Czech Academy of Sciences, Prague, Czech Republic
3   Department of Analytical Chemistry and CAS Key Laboratory of Receptor Research, Shanghai Institute of Materia Medica, Chinese Academy of Sciences, Shanghai, China
4   Biognosys, Zurich-Schlieren, Switzerland
5   European Molecular Biology Laboratory, Heidelberg, Germany
6   Department of Systems Biology, Columbia University, New York, NY, USA
7   Institute for Neuroscience, D-HEST, ETH Zurich, Zurich, Switzerland
8   Statistical Bioinformatics Lab, DMLS, University of Zürich, Zurich, Switzerland
9   Department of Pharmacology, Yale University School of Medicine, New Haven, CT, USA
    *Corresponding author. Tel: +1 203 737 3853; E-mail: yansheng.liu@yale.edu

with stable isotope-labeled amino acids in cells, i.e., pulse SILAC (or pSILAC) technique (Pratt *et al*, 2002; Schwanhausser *et al*, 2009, 2011; Eichelbaum & Krijgsveld, 2014; Jovanovic *et al*, 2015). At the absolute scale, previous pSILAC studies have repeatedly discovered that the protein turnover rate can be influenced by protein abundance, because higher abundant proteins normally tend to be less degraded (Claydon & Beynon, 2012; Liu *et al*, 2017a, 2019). Furthermore, others and we have shown that the relationship between $k_{loss}$ and mRNA concentration is informative in understanding the protein turnover regulation between conditions (Schwanhausser *et al*, 2011; McShane *et al*, 2016; Liu *et al*, 2017a, 2019). By per-gene relative analysis, we revealed that the protein degradation for subunits of heteromeric protein complexes was preferably regulated to buffer against the chromosomal aneuploidy impact due to trisomy 21 or high-grade genomic instability between different HeLa cell strains (Liu *et al*, 2017a, 2019). However, to date, a detailed, systematic investigation of mRNA–$k_{loss}$ correlation has been still lacking. Such an analysis could be fundamental for understanding of the cellular proteotype shaping processes and protein buffering mechanisms, because previous reports have suggested the adaption of translation rates might play a very limited role in buffering of proteins (Albert *et al*, 2014; Bader *et al*, 2015), and that the protein turnover regulation could ensure the robustness of protein expression when transcript levels alter significantly (Stingele *et al*, 2012; Liu *et al*, 2017a).

Furthermore, eukaryotic mRNA alternative splicing (AS) constitutes an important source of protein diversity (Maniatis & Tasic, 2002). It has been reported that most (i.e., ~ 95%) of multi-exon human genes can undergo AS events (Wang *et al*, 2008; Mollet *et al*, 2010). Aberrant AS has been shown to be associated with many diseases (Garcia-Blanco *et al*, 2004; Kahles *et al*, 2018). To study the impact of AS on proteome diversity, several tools and workflows have been developed which convert RNA-Seq data into sample-specific sequence databases to facilitate mass spectrometry (MS)-based identification of the AS-specific peptides (Sheynkman *et al*, 2014; Zhu *et al*, 2014; Tran *et al*, 2017; Wu *et al*, 2019). However, due to the stochasticity and limited sensitivity of MS analysis, as well as the inefficiency of bottom-up strategies in detecting AS proteoforms (Smith & Kelleher, 2013; Wang *et al*, 2018b), determining the overall fraction of biologically relevant AS remains an unsolved problem (Blencowe, 2017; Tress *et al*, 2017a). We have developed an approach differentiating the major and unique AS isoforms for each AS group. Based on the abundance of the major transcript AS isoform, we confirmed that protein abundance is generally correlated to transcript AS levels in a spliceosome-disrupted system (Larochelle, 2017; Liu *et al*, 2017b).

Inspired by the importance of both mRNA–$k_{loss}$ biological relationship and the AS resolution in analyzing mRNA–protein correlation, in this study we investigated the quantitative correlation between mRNA AS levels and proteoform degradation rates. Previously, Zecha *et al* (2018) used a method combining pSILAC and tandem mass tag (TMT) labeling for studying proteoform-resolved protein turnover. However, their survey investigated the absolute level, and no global, relative correlation analysis was performed. Furthermore, in that particular report, the authors did not use any sample-specific proteomic database derived from the corresponding RNA-Seq data to improve AS isoform detection (Zecha *et al*, 2018). Herein, we present an integrative analysis comprising of (i) a sensitive and reproducible data-independent acquisition mass spectrometry (DIA-MS) measurement (Gillet *et al*, 2012; Bruderer *et al*, 2017; Amon

*et al*, 2019; Mehnert *et al*, 2019), (ii) an optimized pSILAC-DIA workflow, (iii) a sample-specific RNA-Seq-derived protein sequence database, and (iv) a novel transcript abundance directed strategy for calling and quantifying AS proteoforms. We then analyzed the protein turnover of differential AS groups across multiple HeLa cell lines collected from different laboratories (Liu *et al*, 2019). Our results at the isoform resolution demonstrate that the mRNA–protein degradation correlation has profound biological implications.

## Results

### An improved pSILAC-DIA-MS workflow for measuring protein degradation

It has been proposed that targeted proteomic approaches will enable the consistent detection of AS-specific peptides across samples (Rost *et al*, 2015; Schreiner *et al*, 2015; Tress *et al*, 2017a). Further, increasing MS analytical sensitivity is essential for identifying new AS-specific peptides (Schreiner *et al*, 2015). Here, we employed an optimized, sensitive DIA-MS method on a high-resolution Orbitrap platform (Bruderer *et al*, 2017; Amon *et al*, 2019; Mehnert *et al*, 2019) and re-measured proteomic samples used in a previous multi-omic study, in which heterogeneity of HeLa cells across different research laboratories was analyzed (Liu *et al*, 2019). These HeLa cell lines were shown to harbor a considerable heterogeneity on mRNA, protein, and protein degradation stemming from copy number variations (CNV) accumulated due to genomic instability and clonal effects (Liu *et al*, 2019). Herein, using the same MS sample sets of HeLa strains, the identical spectral library that contains mass spectrometric assays for 10,000 human proteins (Rosenberger *et al*, 2014), and the same statistical threshold [1% peptide and 1% protein FDR (Rosenberger *et al*, 2017)], we were able to identify a total of 86,996 unique peptides (105,811 peptide precursors) corresponding to 6,552 canonical Swiss-Prot proteins by our new DIA-MS platform with 2-h measurement per sample (Mehnert *et al*, 2019). This result represents 156 and 51% increases of peptide and protein numbers, compared to the previous SWATH-MS results acquired on an earlier instrument (Liu *et al*, 2019). To benefit from the substantially improved sensitivity of this single-shot DIA-MS, we analyzed the samples originating from six HeLa Kyoto strains and six HeLa CCL2 strains to profile both total protein level abundances and the proxy protein degradation rates ($k_{loss}$, by pulsed SILAC labeling, see Materials and Methods) across cell lines (Fig 1A and Appendix Fig S1).

Previously, to analyze pSILAC-DIA-MS data, we used a workflow consisting of following steps (Rost *et al*, 2016; Liu *et al*, 2017a, 2019). First, a spectral library of the "light" isotopic peptides was acquired by shotgun proteomics. Second, the "heavy" version of peptide assays was then generated *in silico*. Both "heavy" and "light" versions were queried independently by OpenSWATH (Rost *et al*, 2014; Navarro *et al*, 2016; Rosenberger *et al*, 2017) to detect and quantify peptides and proteins. Then, the two channels per peptide were linked by the feature alignment algorithm TRIC (Rost *et al*, 2016) in all the samples to acquire the turnover data. Although this workflow substantially increases consistency of quantification (Rost *et al*, 2016), it can be limited if the peak group of one of the channels is below the limit of detection of the OpenSWATH peak picker and may introduce wrongly aligned heavy-to-light pairs.

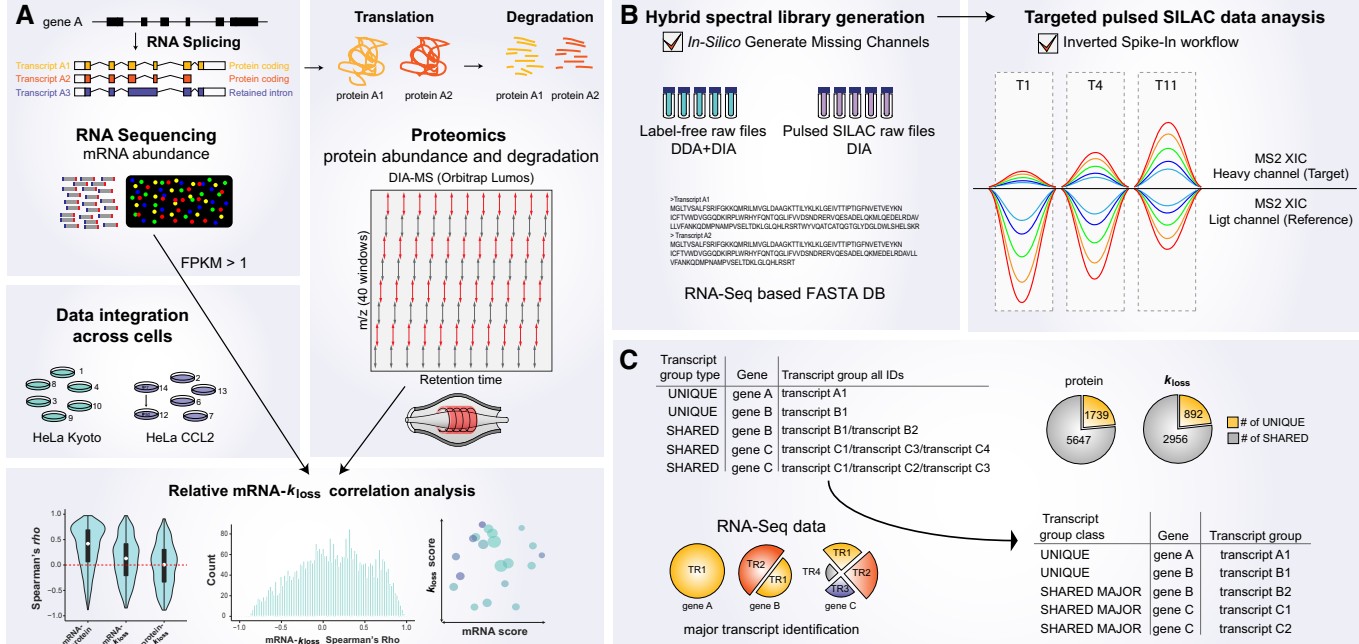

**Figure 1. Experimental and data analysis workflow to study correlation between mRNA abundance and protein level degradation using DIA-MS.**

A  Isoform-resolved protein expression and degradation analysis using DIA-MS. RNA splicing isoforms were analyzed using RNA-Seq, and the total proteome and protein degradation were analyzed using DIA-MS. A protein FASTA database was compiled using protein coding sequences expressing above a threshold of FPKM > 1. After data integration, the splicing isoform-resolved matrix was used to study absolute and relative correlation between mRNA, protein, and protein degradation ($k_{loss}$).

B  pSILAC-DIA workflow for determining protein degradation. A hybrid pSILAC library was created by combining both label-free and labeled DIA- and DDA-MS runs and enabling the *In-Silico* Generate Missing Channels function (in, e.g., Spectronaut). To perform targeted pSILAC data analysis, the Inverted Spike-In workflow (ISW) was used.

C  mRNA abundance directed detection and quantification of protein isoforms and isoform groups. The non-unique peptides were assigned a unique ID by using the average abundance on mRNA level for all proteins included in a protein group. The major (i.e., the most abundant) splicing isoform on mRNA level was selected as the best representative ID for the protein group (*shared major*).

To compensate for the limitations, we herein adopted the peptide identification strategy used in the inverted spike-in workflow (ISW) (Reiter *et al*, 2011) for analyzing pSILAC-DIA data (Fig 1B). ISW was initially introduced in a particular type of SRM experiments where the "light" synthetic peptides were spiked as standards. This means, in ISW the peptide detection scoring process is only based on the *q*-values of "light" precursors. Thus, in pSILAC-DIA dataset ISW will maximize reliable detection of newly synthesized "heavy" protein in the early time points during labeling, when the "heavy" signals are much lower than those "light" ones of pre-existing protein copies. Indeed, ISW increases the number of heavy-to-light ratios by 30.6 and a 14.7% in the dataset with 1- and 4.5-h pSILAC labeling (see Appendix; Reiter *et al*, 2011). The ISW is now available in a new version of Spectronaut software (Bruderer *et al*, 2015) by which both "heavy" and "light" MS assays can be reversibly generated based on either DIA or DDA datasets or both. Furthermore, the heavy-versus-light elution was assembled before DIA identification, so that no post-feature alignment (Rost *et al*, 2016) is required (see Appendix for a step-by-step protocol for pSILAC data analysis and related data assessment).

We have compared pSILAC-DIA to two alternative workflows, pSILAC-MS1 (Schwanhausser *et al*, 2011; Mathieson *et al*, 2018) and pSILAC-TMT (e.g., when combined with synchronous precursor selection (SPS)-based MS3 scanning for improving the TMT quantification, see Appendix; McAlister *et al*, 2014; Welle *et al*, 2016;

Savitski *et al*, 2018; Zecha *et al*, 2018). We identified particular advantages of pSILAC-DIA over the existing methods, such as lower charge states of MS2 signals which reduced MS data processing difficulty (Appendix Fig S2), significantly better accuracy in quantifying SILAC heavy-to-light (H/L) ratios than pSILAC-MS1 (whose performance was shown to be comparable to pSILAC-TMT (Zecha *et al*, 2018); Appendix Figs S3 and S4), much more quantitative data points with the potential to use most high-resolution heavy and light fragment ions (Appendix Fig S5) together with their elution traces along the liquid chromatography (Appendix Fig S6), as well as other flexibilities and potentials. Just as an example, by analyzing the protein level pSILAC data at 1, 4.5, and 11 h (T1, T4, T11) for HeLa 7, we found the MS2-derived H/L ratios in DIA-MS have a much narrower distribution than the MS1 data in the same runs (Fig 2A). The top 6 most abundant fragment ions yielded 825,041 H/L pairs, which could be used to further increase the quantitative accuracy by interference filtering [to 450,407 pairs, using selection algorithms provided by Spectronaut or other software such as mapDIA (Teo *et al*, 2015)] and by summarization at the peptide precursor level (Fig 2B). Figure 2C–F illustrated a protein example of endoplasmic reticulum chaperone BiP (HSPA5). This example demonstrates that, conceivably, more MS2 H/L pairs are obtained from pSILAC-DIA, compared to the limited quantitative features obtained from pSILAC-MS1 (based on the precursor pairs) or from pSILAC-TMT (based on the reporter ion ratios in identified MS2 or SPS-MS3

scans). Considering other potentials of pSILAC-DIA such as resolving post-translational modifications, low cost, and experimental design flexibility (see Appendix), we conclude that pSILAC-DIA is a competitive and promising approach for determining protein turnover in various systems.

In summary, we optimized a sensitive, accurate, and reproducible pSILAC-DIA workflow by the development of both a technical platform and a data analysis strategy.

## mRNA abundance directed detection and quantification of protein AS isoforms and isoform groups

To further increase the coverage of AS isoforms by proteomics, we devised a novel heuristic strategy for mapping the isoforms. Owning to the dynamic range of a single-shot MS analysis in cell lines (Beck *et al*, 2011; Ebhardt *et al*, 2012), we reason that our MS measurement cannot detect a protein or a proteoform that has an extremely low mRNA abundance. To corroborate, we plotted the mRNA–protein absolute correlation based on FPKM values and DIA-MS intensities (Fig EV1A). We found that only 0.74% of detectable unique peptides (i.e., 83 out of 11,234 peptide full profiles; test of 14 DIA-MS samples) seemed to be translated from transcripts of average FPKM < 1. Therefore, we compiled a FASTA protein sequence database (DB) for MS

data analysis using translated sequences from AS isoforms expressing at FPKM > 1 in at least three HeLa cell lines (Nagaraj *et al*, 2011; Hart *et al*, 2013; Fig EV1C). Accordingly, only 11,409 genes, instead of the whole genome, were considered to be possibly detected by MS. Using this simple cutoff, we found that those peptides uniquely mapping to only one splicing form, i.e., "unique hits" or "proteotypic" (Mallick *et al*, 2006), successfully increased by 35.22% as compared to the results with no FPKM cutoff (Fig EV1D). Furthermore, the cutoff of FPKM > 1 yielded a FASTA DB with 3.23 isoforms per gene and the detection of 1.87 isoforms per gene, as compared to 2.08 and 1.61 isoforms per gene when the non-canonical UniProtKB DB was used (i.e., a scenario without a sample-specific RNA-Seq dataset; Fig EV1F and G; Zecha *et al*, 2018). Finally, the FPKM > 1 cutoff enabled detection of 47,091 peptide precursors whereas FPKM > 0 (Fig EV1B) detected 46,625 (Fig EV1E and H), indicating there is no compromise of peptide detection for FPKM > 1 cutoff. Collectively, in all 12 HeLa cell lines, we quantified the abundance of 51,291 peptide full profiles and 24,275 peptide $k_{loss}$ values (see Materials and Methods). This translates to 6,994 protein AS groups (3,518 gene symbols collapsed) quantified with a degradation rate in every of the 12 HeLa samples (Table EV1). Among these, 15,671 peptides and 6,890 peptide $k_{loss}$ values were found to be unique hits. Thus, the mRNA abundance information was used to assist the protein AS detection.

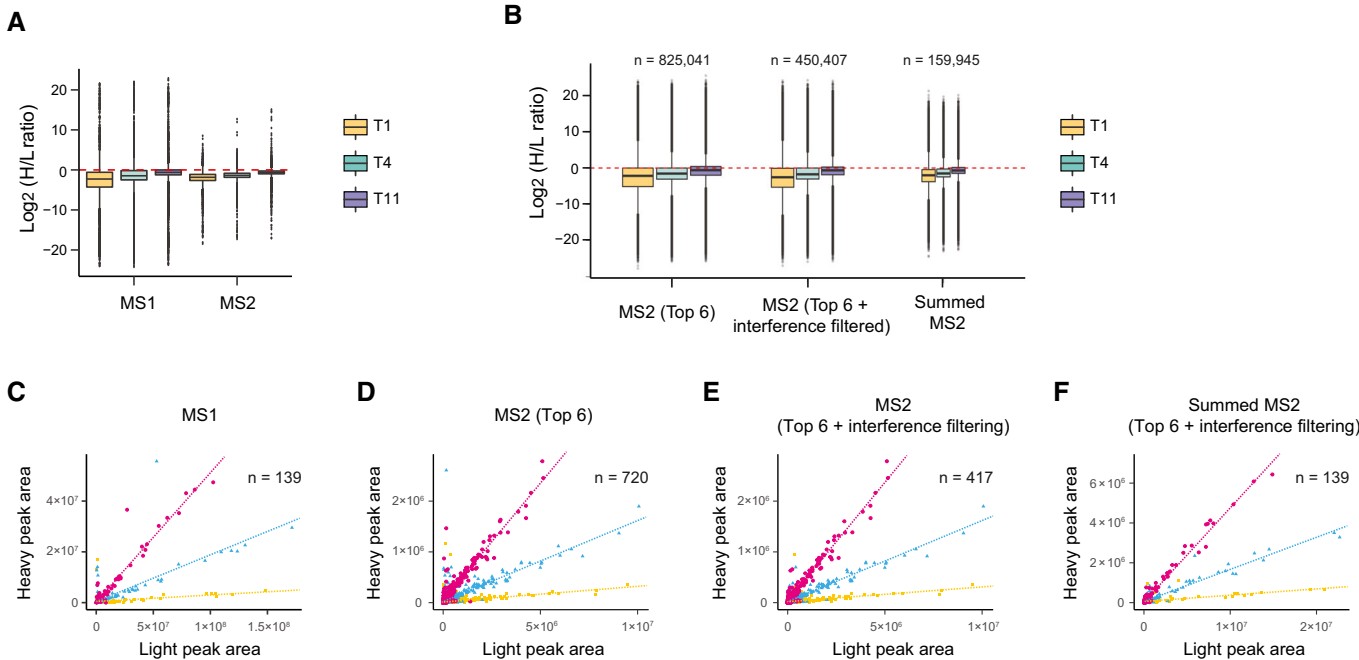

**Figure 2. The MS2-based pSILAC-DIA improved the quantification accuracy determining Heavy-to-Light (H/L) ratios and increased the number of quantitative features, as compared to MS1-based approach.**

A    H/L ratio distributions of all protein AS groups are shown at the 1, 4.5, and 11 h (T1, T4, T11) for a randomly selected HeLa cell line (HeLa 7). The median for each protein AS group was calculated using all peptide precursors with MS1- and MS2-derived ratios.

B    Distributions of all MS2-derived H/L ratios. The fragment ion H/L ratios were calculated using the Top 6 fragment ions or Top 6 fragment ions (left boxes) from which the potentially interfering ions can be removed (middle boxes), and the peptide precursor H/L ratios can be summarized using the filtered fragment ions (right boxes). The numbers of quantitative data features in each time point are indicated; the width of the boxplot is scaled to the number of values. Box borders represent the 25th and 75th percentiles, bar within the box represents the median, and whiskers represent the minimum and maximum value within 1.5 times of interquartile range. The dashed red line indicates zero.

C–F   Quantitative features for endoplasmic reticulum chaperone BiP (HSPA5) as an example. The correlations between SILAC light and heavy intensities of the Top 6 (D), Top 6 filtered for interfering ions (E), summed MS2 (F), and MS1 (C). A linear regression line (dashed line) was added for each time point indicated by different colors. *N* denotes the number of quantitative values for respective signals.

Next, to assign quantitative information for more protein AS groups, we conducted an inference strategy similar to a previous approach (Liu *et al*, 2017b). Briefly, after quantifying "unique hits" (UQ), we retrieved average mRNA abundance across cells for each splicing isoform from the same gene included in a shared protein group (Fig 1C). The major (i.e., the most abundant) splicing isoform on mRNA level was then selected as the best representative proxy for the whole protein AS group ("shared major", or SM proteins). This assignment is based on the observation that lower abundant transcripts are less likely to manifest on protein level (Fig EV1A; Liu *et al*, 2017b), and that most protein coding genes have one major transcript expressed at significantly higher levels than the others (Gonzalez-Porta *et al*, 2013; Abascal *et al*, 2015; Blencowe, 2017; Tress *et al*, 2017a). Totally, by UQ mapping we successfully quantified the expression of 1,739 proteins and calculated $k_{loss}$ values for 892 unique proteins. By SM mapping, we quantified 5,647 protein AS groups and turnover rates for 2,956. Thus, the mRNA abundance was also used to assign the AS quantities at the proteomic level.

The $k_{loss}$ values, respectively, retrieved from UQ and SM harbor comparable values, which classified the 12 HeLa cell lines into two clusters, CCL2 and Kyoto (Fig 3A and B). Also, the protein Kyoto/CCL2 fold changes derived from UQ and SM peptides strongly correlated to each other for the same proteins ($\rho = 0.88$; Fig 3C). To estimate the impact of UQ and SM assignments for correlation analysis, we inspected the Spearman's correlation between mRNA, protein, and $k_{loss}$ values across-proteins (i.e., absolute correlation; Fig 3D and E) and between conditions (i.e., relative correlation, using fold changes between six Kyoto and six CCL2 cell lines; Fig 3F and G). We confirmed that all the correlations obtained from both UQ and SM categories were very similar and that all the correlation trends were consistent to or even slightly better than our previous gene-centric report (Liu *et al*, 2019; Fig 3D–G). Thus, both UQ and SM categories are used in following up analysis.

In summary, we adopted an mRNA abundance directed heuristic approach (Lau *et al*, 2019), ensuring the downstream functional analysis can be performed at AS isoform resolution.

### Absolute and relative $k_{loss}$ comparisons

The HeLa Kyoto and CCL2 cells were demonstrated to be fundamentally distinctive in both genome and proteome (Liu *et al*, 2019), presenting a strong case for relative comparison. Extending on these observations, we then investigated whether relative $k_{loss}$ regulation could provide particular biological insights, based on the new isoform-specific dataset.

Our first goal was to assess the role of protein degradation in protein abundance buffering (Mueller *et al*, 2015; Liu *et al*, 2016; Wang *et al*, 2018a). The absolute mRNA–$k_{loss}$ correlation was determined to be $\rho = -0.14$ and $-0.17$ (Fig 3D and E; $n = 885$ and 2,895, $P < 0.001$) in UQ and SM proteins, and the absolute protein-$k_{loss}$ correlation is even more negative ($\rho = -0.31$ for UQ and $-0.29$ for SM). Thus, protein degradation seems to be slower for higher gene expression. However, when the Kyoto/CCL2 fold changes are analyzed, slight but significant positive across-gene mRNA–$k_{loss}$ correlations can be obtained (Fig 3F and G; $n = 885$, $\rho = 0.08$, $P = 0.017$ for UQ; $n = 2,895$, $\rho = 0.10$, $P < 0.0001$ for SM). Hence, the absolute correlation analysis can be misleading in this case due to the confounding factor of high abundant proteins having a tendency to

carry housekeeping functions, which are thus prone to slower degradation (Claydon & Beynon, 2012; Liu *et al*, 2017a, 2019). Therefore, by relative correlation of mRNA–$k_{loss}$, we can confirm that protein turnover might be more likely used by the cells as an attempt to buffer against, rather than to reinforce, the mRNA variation between conditions. This conclusion was verified by another modeling algorithm that only analyzes the light-channel, unlabeled peptides during pSILAC labeling for estimating protein degradation (see Appendix).

*Secondly,* we investigated the proportion of proteome significantly regulated by protein degradation. We summarized the isoform-resolved protein-specific Spearman's $\rho$ across 12 HeLa cell lines (Fig EV2A and B) and further classified the proteins based on whether there were any significant changes of Kyoto versus CCL2 at mRNA, protein, or protein degradation levels (Fig 3H–J). The mRNA–$k_{loss}$ correlation was increased for protein AS isoforms that were not differentially expressed (median $\rho = 0.18$ and 0.16 UQ and SM; Fig 3H) and was significantly lower for proteins differentially expressed (median $\rho = 0.04$ and 0.08; Fig 3H), indicating that $k_{loss}$-based buffering associates with lower protein level variation between states. Correspondingly, the mRNA–protein correlation is high for differentially expressed proteins (median $\rho = 0.70$ and 0.66, UQ and SM; Fig 3H), indicating protein level variability globally follow mRNA changes. Interestingly, only 10.96% (i.e., 97 out of 885) and 12.61% (i.e., 365 out of 2,895) of UQ and SM proteins had a markedly regulated degradation rate (Fig 3J).

*Thirdly,* we asked if the relative $k_{loss}$ differences are associated with different GO biological processes (GOBP) and cellular components (GOCC) than absolute $k_{loss}$ comparison. Consistent to Zecha *et al* (2018), we discovered that different GOBPs and GOCCs are degraded with different speed at the absolute scale (Fig 4A and B, and Tables EV2 and EV3). For example, "mitochondrial large ribosomal subunit" ($P = 0.004$), "condensed chromosome kinetochore" ($P = 0.01$), and "plasma membrane" ($P = 0.002$) were significantly enriched among the groups with fast degradation rate. The corresponding GOBPs are "mitochondrial translational elongation" ($P = 0.001$), "cell division" ($P < 0.0001$), and "cell migration" ($P = 0.001$). On the other hand, cellular components "peroxisome" ($P = 0.0003$), "endoplasmic reticulum lumen" ($P = 0.0003$), and "mitochondrial matrix" ($P < 0.0001$) and relevant GOBPs were degraded at much slower rates. Notably, however, the relative difference of $k_{loss}$ between HeLa CCL2 and Kyoto enriched distinctive GOBPs and GOCCs (Fig 4C and D, and Tables EV4 and EV5; $P < 0.01$). To elaborate, we select different mitochondrial compartments as an example (Fig EV3): "Mitochondrial matrix" was significantly enriched in "slow degradation", whereas "mitochondrial respiratory chain complex I" was enriched in "fast degradation" ($P < 0.0001$; Fig EV3A). Faster degradation of the complex I components was proposed to be caused by the high local oxidative stress (Zecha *et al*, 2018). However, the "complex I" proteins are much less abundant than "matrix" proteins. Thus, the dramatic degradation rate difference between the two might be also ascribed to the absolute protein abundance difference (Fig EV3B) because of the general trend that highly expressed genes have a lower $k_{loss}$. In another example, both "mitochondrial large ribosomal subunit" and "complex I" are both low-abundant protein groups (Fig EV3B), but showed distinctive, up- and down-regulated protein degradation between HeLa Kyoto and CCL2 cells ($P < 0.001$; Fig EV3C).

*Fourthly,* we found that relative $k_{loss}$ highlights different individual proteoform degradation and expressions than the absolute

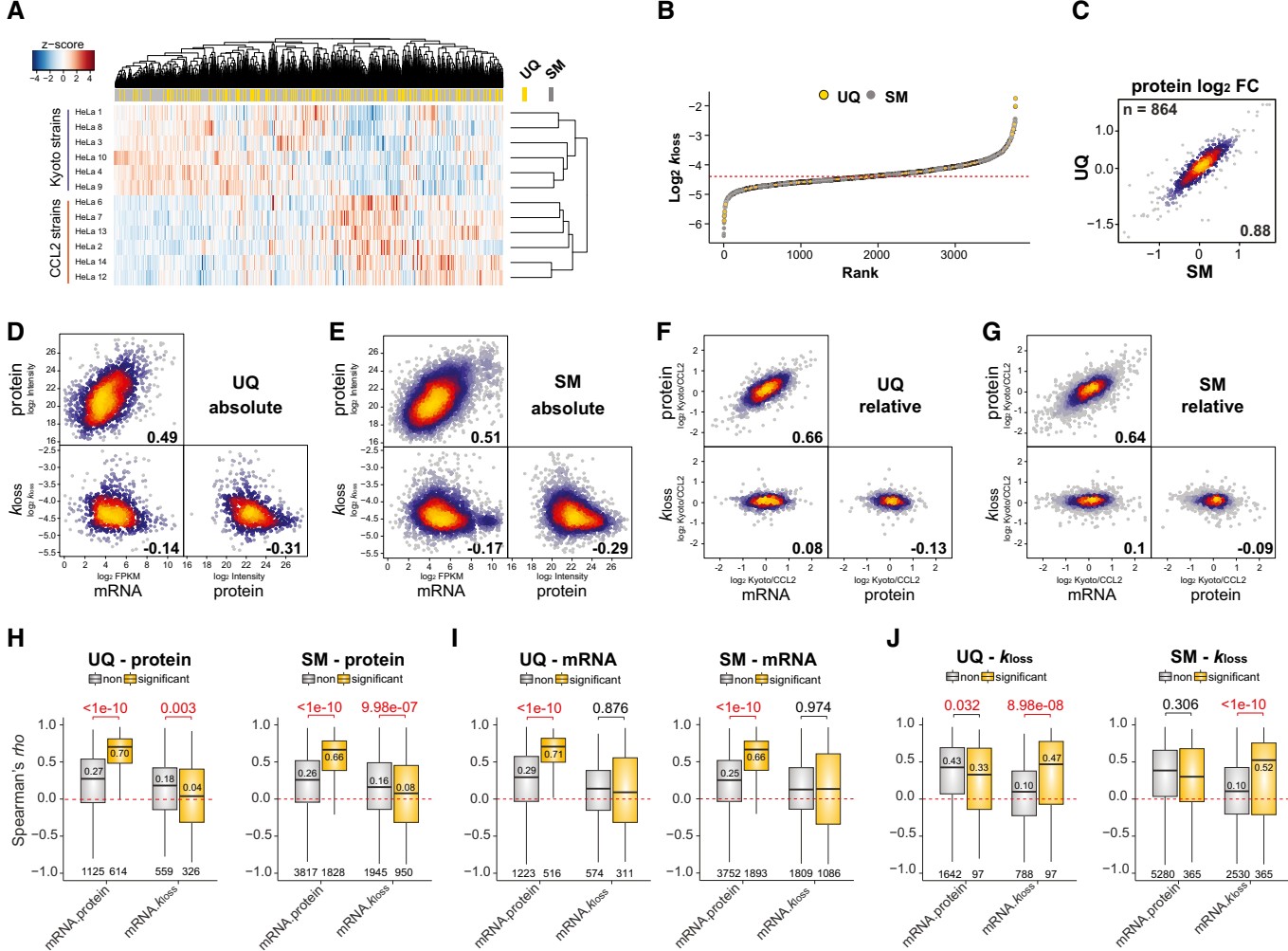

**Figure 3. Absolute and relative correlation analysis in a splicing isoform-resolved data set.**

A   Hierarchical clustering of the $k_{loss}$ values quantified for twelve HeLa cell lines. The positions of unique (UQ) and shared major (SM) proteins were highlighted in the separated bar.

B   Rank distribution of the averaged $k_{loss}$ values of UQ and SM proteins across HeLa cells. The dashed red line indicates median.

C   Correlation of Kyoto/CCL2 protein fold change for IDs quantified in both UQ and SM.

D, E   Across-gene Spearman's absolute correlation between indicated values (average from all cell lines).

F, G   Spearman's correlation between indicated values using relative quantification data (i.e., Kyoto/CCL2 fold change).

H–J   Protein AS group-specific Spearman's rho distribution of isoforms that were differentially expressed on mRNA level (I, EdgeR, adjusted $P < 0.01$), protein level (H, $t$-test, adjusted $P < 0.01$), or degraded (J, $t$-test, adjusted $P < 0.01$) between Kyoto and CCL2 (yellow) and which are not (gray). The numbers indicate Wilcoxon $P$-values (top), median rho of significant comparisons (above or below the median bar), and number of values (bottom). Spearman's rho was calculated for every protein AS group in the data set using twelve data points and summarized across genes/proteins. Box borders represent the 25th and 75th percentiles, bar within the box represents the median, and whiskers represent the minimum and maximum value within 1.5 times of interquartile range. The dashed red line indicates zero.

$k_{loss}$. Using statistics (Student's $t$-test $P < 0.05$ and mean $\log_2$ fold difference > 0.32) and stringent criteria (two unique peptides per protein AS group), we discovered 47 genes for which we revealed significant $k_{loss}$ difference between peptides originating from different AS isoform groups in both HeLa CCL2 and Kyoto cells (examples in Fig 4E). Moreover, the peptide intensities indicating proteoform-level expression followed opposite trends (Fig 4F). We then asked whether the splicing isoforms could be differentially degraded relatively between Kyoto and CCL2 cells. Several genes were found to exhibit such a behavior (examples in Fig 4G; $t$-test

or ANOVA $P < 0.05$ and mean $\log_2$ difference > 0.32). However, many of the corresponding peptide intensities fold changes do not show any reciprocal trend to $k_{loss}$ in the relative comparison (Fig 4H), consistent to Fig 3F and G. The across-sample variation of these peptides within AS isoforms is smaller than that between AS isoforms, demonstrating the biological significance (Appendix Fig S7C). Thus, the degradation variability can be observed at the individual AS isoform level in both absolute and relative scale, but the relative result seems to be independent of peptide intensities.

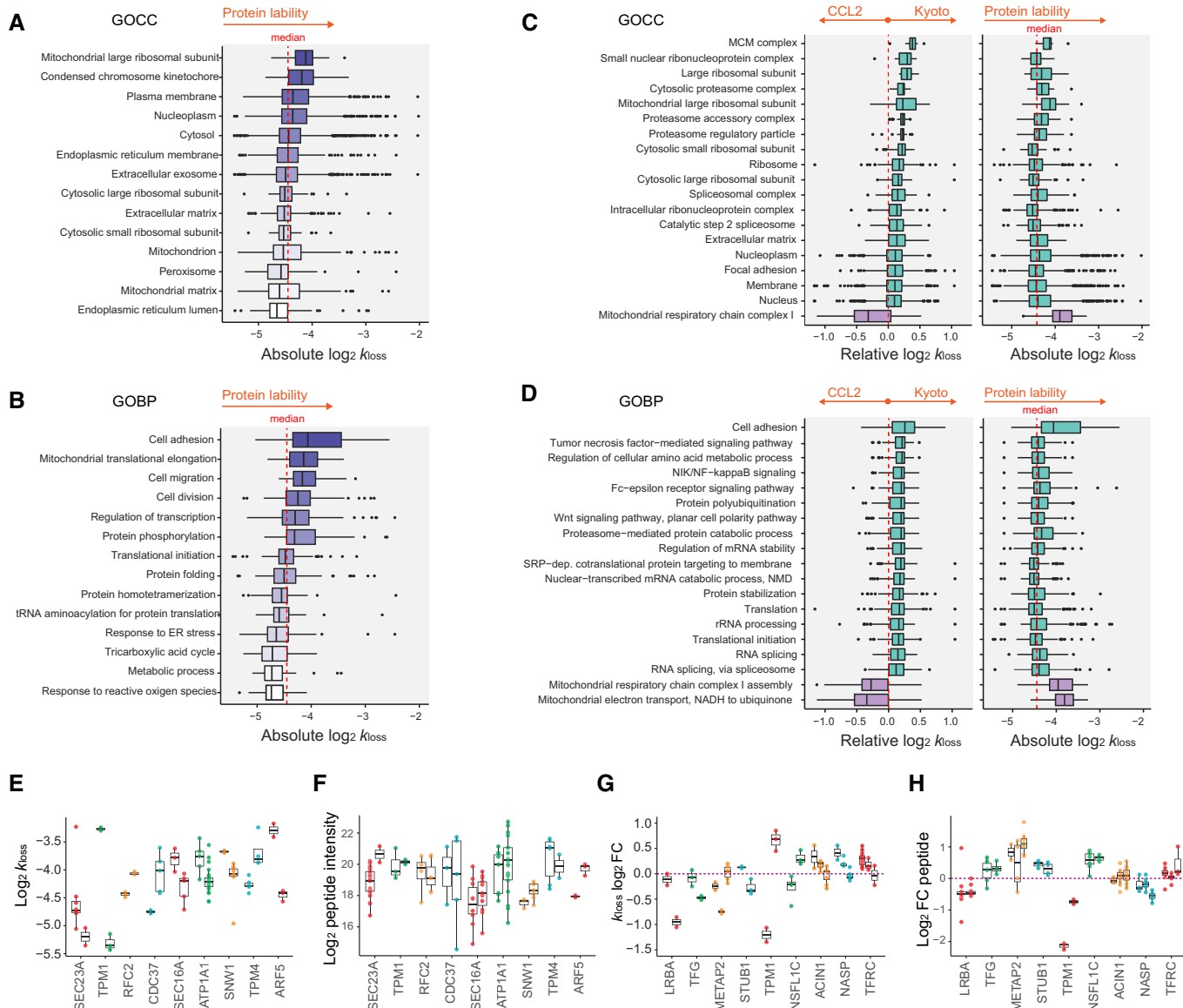

**Figure 4. Differential functional categories and isoforms revealed by enrichment analysis of absolute and relative $k_{loss}$ measurements.**

A, B   Distribution of absolute $k_{loss}$ values (HeLa average) as a function of cellular localization (A, GOCC, n = 29, 23, 325, 770, 889, 146, 798, 49, 82, 35, 393, 25, 127, and 44, from top to bottom) or biological functions (B, GOBP, n = 34, 49, 23, 82, 128, 36, 114, 93, 25, 29, 22, 21, 35, and 17, from top to bottom); the color intensity gradient highlights the increase in protein degradation rate.

C, D   Differential degradation ($k_{loss}$ $\log_2$ FC) of GOCC (C, n = 7, 10, 9, 7, 29, 16, 11, 35, 113, 49, 52,  84, 60, 82, 770, 179, 643, 936, and 14, from top to bottom) and GOBP (D, n = 34, 40, 37, 41, 47, 52, 43, 60, 64, 79, 86, 37, 135, 148, 114, 75, and 132, from top to bottom) between HeLa Kyoto and CCL2 (1D enrichment test, P < 0.05). The second panel shows absolute $\log_2$ $k_{loss}$ values matched to the selected examples.

E   Absolute peptide $k_{loss}$ (CCL2 average) distribution of differentially degraded AS isoforms of the same gene (t-test P < 0.05, mean $\log_2$ FC > 0.32).

F   Absolute peptide intensities of AS isoforms matching to (E).

G   Relative peptide $k_{loss}$ (Kyoto/CCL2 FC) distribution of differentially degraded AS isoforms of the same gene (t-test or ANOVA P < 0.05, mean $\log_2$ fold difference > 0.32, Tukey test for pairwise comparisons).

H   Distribution of HeLa Kyoto/CCL2 peptide intensity $\log_2$ FC matching to examples in (G). The number of values is represented by the number of dots.

Data information: In (A–H), the box borders represent the 25th and 75th percentiles, bar within the box represents the median, and whiskers represent the minimum and maximum value within 1.5 times of interquartile range. The dashed red line indicates zero or median as indicated.

## Biological annotation of relative mRNA–$k_{loss}$ correlation

From the distribution of $k_{loss}$ ratios between HeLa Kyoto and CCL2 strains shown above (Figs 3 and 4), protein degradation can be employed by cells trying to adjust gene expression. Intriguingly,

irrespective of translational regulation and protein abundance (Appendix Fig S8), higher post-translational regulation effort (i.e., higher mRNA–$k_{loss}$ correlation), generally succeeded in reducing the protein concentration variability between cells: the top 20% proteins (Q5) with highest mRNA–$k_{loss}$ correlation show lower coefficients of

variation (CV) than the bottom 20% proteins (Q1), and both groups were significantly different from the rest of the data (Q2–Q4), confirming the modulation of protein levels by protein degradation-mediated buffering mechanisms (Kruskal–Wallis $P < 1e\text{-}10$; Fig 5D). This is consistent to previous reports, suggesting that the adaption of translation rates does not or, only partially explain buffering of proteins (Albert *et al*, 2014; Bader *et al*, 2015).

To illustrate the diversity of protein degradation between cell lines, we assessed the distribution of the mRNA–$k_{loss}$ correlation summarized for each protein. As shown in Fig 5A, the protein AS group-specific mRNA–$k_{loss}$ Spearman's ρ was widely distributed between −1 (indicating a strong negative correlation) and 1 (indicating a strong positive correlation). The distribution is skewed toward positive values [median ρ = 0.13: There are 688 protein AS groups showing a strong positive correlation (ρ > 0.576, $P < 0.05$, purple) and only 242 protein AS groups showing a strong negative regulation (ρ < −0.576, $P < 0.05$, green)]. This skewed distribution again indicates mRNA regulation attempts are more often prone to be buffered rather than to be amplified by protein degradation.

Selected examples of GOBPs and GOCCs were shown to be strongly enriched in positive, insignificant, or negative mRNA–$k_{loss}$ correlation ranges (Tables EV6 and EV7). While biological processes such as "mRNA splicing" and "translation" showed a significant trend toward positive correlation (median ρ = 0.283 and 0.336, $P < 0.0001$), "protein catabolic process" and "establishment of protein localization" represent GOBPs in which mRNA and $k_{loss}$ tend to anti-correlate (median ρ = −0.325 and −0.168, $P < 0.02$). On the other hand, "carbohydrate metabolic process" and "cell adhesion" represent GOBPs with no clear trend of correlation (median ρ = −0.04 and 0.002; Fig 5B). Likewise, proteins in GOCCs such as "spliceosomal complex", "ribosome", and "nuclear chromosome" tend to have positive mRNA–$k_{loss}$ relative correlation (median ρ of 0.379, 0.357, and 0.341, $P < 0.002$), whereas "cell–cell junction" and "membrane raft" showed a negative correlation (median ρ of −0.171 and −0.157, $P < 0.05$; Fig 5C).

We further visualized the relationship between mRNA and $k_{loss}$ by performing a two-dimensional enrichment analysis defined by relative scores of mRNA (*x*-axis) and $k_{loss}$ fold changes (*y*-axis) between HeLa Kyoto and CCL2 (Cox & Mann, 2012; Tyanova *et al*, 2016). In this 2D space, the position of a functional category (Tables EV8 and EV9) essentially indicates the trends of protein degradation under mRNA up- or down-regulations (Fig 5E and F). For instance, GOBPs (Fig 5E) and GOCCs (Fig 5F) related to mRNA splicing

(e.g., "spliceosomal snRNP assembly", "mRNA splicing", and "spliceosomal complex"), protein synthesis ("translation" and "ribosome"), or mitochondria subunits ("mitochondrial large ribosomal subunit" and "mitochondrial nucleoid") were mostly localized in upper right quadrant, indicating the buffering attempt against their mRNA changes through protein degradation. On the other hand, a few terms such as "cell adhesion" or "protein catabolic process" are located in the upper left quadrant, indicating many members of them are under concordant regulation through both transcription and protein degradation. Although many of these GOBPs and GOCCs were identified as differentially degraded between cells (Fig 4), it is important to place the values in the context with mRNA abundance differences to identify the buffering or concerting attempts of protein degradation.

Somewhat surprisingly, we discovered distinctive degradation behavior for different sub-organelles of the same GOCC. The first example is the proteasome (Fig 5G). The mRNA–$k_{loss}$ correlation coefficients of members in KEGG item "Proteasome" uncovered a dramatic difference between the 19S Regulatory (corresponds to GOCC "proteasome accessory complex") and 20S Core particles (corresponds to GOCC "proteasome core complex") (Fig 5I; $P = 0.0149$, Wilcoxon test). In particular, the 20S proteins are preferably degraded to impair the transcript variation (median ρ = 0.416 for mRNA–$k_{loss}$ correlation, as compared to median ρ = 0.028 for 19S proteins). On the other hand, the difference of relative mRNA–protein correlations was found to be insignificant for the protein members of the two subunits ($P = 0.9444$, Wilcoxon test). The relative mRNA–protein correlation is low for both subunits (median ρ of −0.007 and 0.053 for 20S and 19S, respectively). These results indicate strong translational regulation for proteasome 19S, as well as substantial control for 20S proteins at both translational and post-translational levels. Further dissection of 19S particles yielded similar insights (Fig EV4A–D).

Another case is mitochondrial ribosome (or mitoribosome; Fig 5H). Both small and large subunits of the mitoribosome showed a positive mRNA enrichment score indicating relative transcriptional upregulation in Kyoto (Fig 5F), but the directions of $k_{loss}$ fold change were converse for the two subunits. The protein degradation carries a clear trend to buffer transcript variation for large mitoribosome subunit (median ρ = 0.532, mRNA–$k_{loss}$ correlation), but not for the small one (median ρ = −0.014, Wilcoxon test, $P = 0.0022$, Fig 5J). Additionally, statistics revealed that small ribosomal subunit of mitoribosome had a significantly greater mRNA–protein correlation than

**Figure 5. Biological annotation of the isoform-resolved protein-specific mRNA–$k_{loss}$ correlation at the organellar and sub-organellar levels.**

A   Distribution of isoform-resolved protein-specific mRNA–$k_{loss}$ Spearman's *rho* (median ρ = 0.13) across the proteome. The colors highlight significant correlation values ($P < 0.05$). Spearman's *rho* was calculated for every protein AS group in the data set using twelve data points. All *rho* values were then summarized as histograms.

B, C   Selected biological processes (B, GOBP) and cellular compartments (C, GOCC) showing different distribution of mRNA–$k_{loss}$ correlation.

D   The quantitative CVs of all protein AS groups across HeLa cell lines are distributed in five mRNA–$k_{loss}$ correlation segments (Q1–Q5, each represents 20% of the data, include protein AS groups with the lowest and the greatest correlation, respectively; Kruskal–Wallis test $P$-value is shown, pairwise comparisons were performed using pairwise Wilcoxon test with Benjamini–Hochberg correction). Box borders represent the 25th and 75th percentiles, bar within the box represents the median, and whiskers represent the minimum and maximum value within 1.5 times of interquartile range.

E, F   Two-dimensional (2D) enrichment plot of selected GOBPs or GOCCs. The axes denote enrichment score for mRNA expression (*x*) and protein degradation (*y*) log$_2$ FC HeLa Kyoto/CCL2.

G, H   Schematic representation of human proteasome (G, "KEGG: Proteasome") and mitochondrial ribosome (H, GOCC DIRECT); colors correspond to mRNA–$k_{loss}$ correlation (*rho*).

I, J   Statistical analysis of the differences between the proteasome (I) and mitoribosome (J) subunits (indicated by Wilcoxon test $P$-values). The red lines denote median with interquartile range. The number of protein AS isoform groups is indicated by the number of dots.

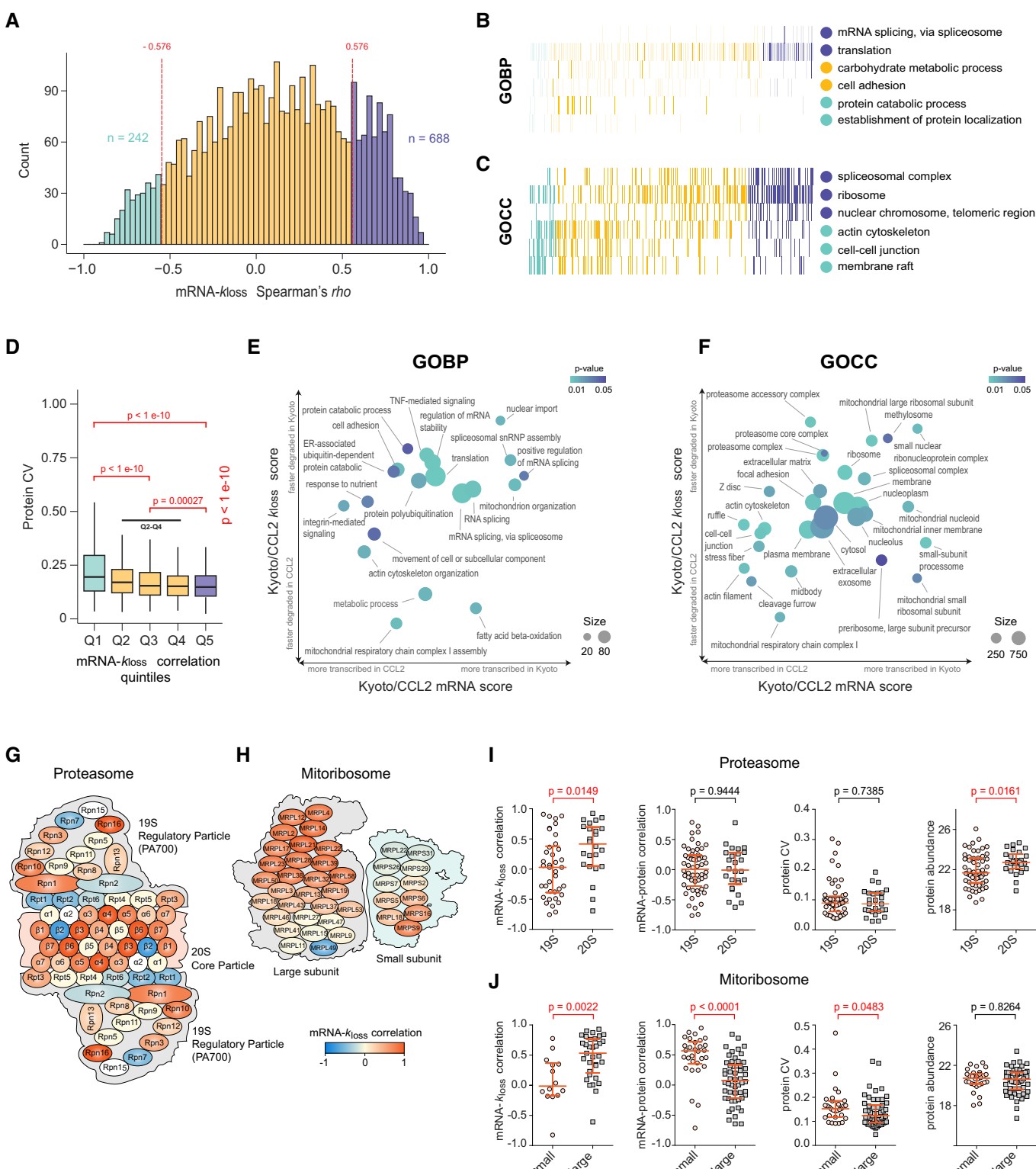

**Figure 5.**

large subunit (median ρ of 0.566 versus 0.073, $P < 0.0001$) and a significantly greater protein CV ($P = 0.0483$). Thus, the protein components of the large subunit of the mitoribosome were buffered by degradation to maintain the protein abundance, and this regulation

significantly reduced the mRNA–protein correlation, while the small subunit protein variation is principally determined by mRNA changes.

In summary, the annotation of relative mRNA–$k_{loss}$ correlation revealed diverse buffering or concerting functions exerted by protein

turnover in different biological processes, organelles, and even in sub-organelle components.

## Intron retention and transcript-level isoform switch events impact protein expression but not degradation

In the HeLa cell line panel, there are two HeLa cell variants that were derived from the same ATCC HeLa CCL2, but harvested at the 7[th] (P7) and 50[th] passages (P50), corresponding roughly to 3 months of passaging. We previously found that 6–7% of genes changed their expression, which also manifested at the protein level. These changes could be attributed to genomic instability due to clonal selection or adaptation in the cell culture conditions (Liu et al, 2019).

The present data uniquely enabled us to address whether AS isoforms generated during P7-to-P50 passaging can contribute to the proteome and protein degradation levels. To do so, we first examined the expression differences of the spliceosome components. We found that the entire spliceosome expression was significantly upregulated at both mRNA and protein levels in P50 (Wilcoxon test $P < 0.0001$; Fig 6A and B). It was previously reported that a depletion of the splicing core component PRPF8 induced intron retention; i.e., the transcript with retained introns (RI) may not be translated but is retained in the nucleus (Wickramasinghe et al, 2015). The RI-dependent mRNA degradation by the non-sense-mediated decay (NMD) (Braunschweig et al, 2014) thus reduced protein levels, which can be detected by DIA-MS (Liu et al, 2017b). The spliceosome upregulation in P50 (as opposed to PRPF8 depletion) might in fact switch a number of RIs to coding events during the cell culturing over the 3 months. We then sought to identify AS switch events (i.e., the most abundant splicing isoform changed between conditions) for P7-to-P50 on the

transcriptome level and to map these events to the protein level. Using a tool called SwitchSeq (preprint: Gonzàlez-Porta & Brazma, 2014; Liu et al, 2017b), we identified 900 switch events between HeLa P7 and P50. As depicted in Fig 6C, transcripts that underwent switching from a RI in P7 to protein coding in P50 (i.e., RI to coding) showed a significant increase on protein level in P50 ($n = 77$, Wilcoxon test, $P = 0.0071$; Figs 6C and EV5D). Switch events from a protein coding isoform in P7 to a different protein coding isoform in P50 (i.e., coding to coding) were used as a control for this comparison. Thus, the loss of RI tends to increase protein expression. In contrast, the protein degradation seems to be independent of, or insignificantly affected by RI events in the presented data (Fig 6D). A representative example is the RI-to-coding switch for PROSC (pyridoxal phosphate binding protein) for which the increase of the protein coding AS during P7-to-P50 transition was quantified (Fig EV5A). Finally, we also detected the case of routine AS switch at the protein level. Two examples are shown in Fig EV5B and C: In the case of GLTSCR2 (ribosome biogenesis protein NOP53), the major protein coding isoform in P50 was upregulated, while for SNAP29 (synaptosomal-associated protein 29), the major isoform in P7 was downregulated during the P7-to-P50 transition. Altogether, the AS switching, especially loss of intron retention, seems to have marginal or minimal effects on the protein level degradation.

## Discussion

The importance of protein turnover has been recognized 80 years ago (Hinkson & Elias, 2011). The data generated by pSILAC experiments (Pratt et al, 2002; Schwanhausser et al, 2009, 2011) helped in discovering how cells adjust proteostasis process (Eichelbaum & Krijgsveld,

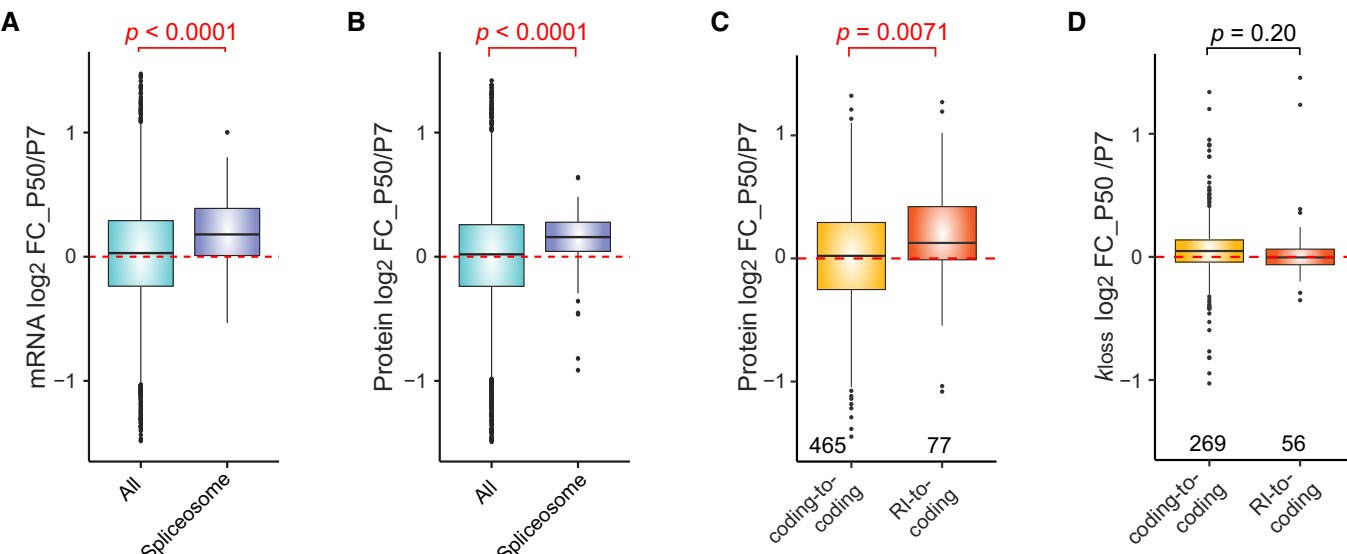

**Figure 6. The impact of alternative splicing and intron retention on protein expression and degradation.**

A, B Distribution of HeLa CCL2 P50/P7 mRNA (A) and protein (B) log$_2$ FC for all IDs (green; $n = 7{,}386$) and components of spliceosome ("KEGG: Spliceosome"; purple; $n = 224$). The Wilcoxon test $P$-values are shown.

C, D Distribution of protein log$_2$ FC (C) and $k_{loss}$ log$_2$ FC (D) between HeLa CCL2 P50 and P7 for proteins translated from AS isoforms undergoing a switch event. The number indicates Wilcoxon test $P$-values (top) and the numbers of values (bottom). RI, retained intron.

Data information: In (A–D), the box borders represent the 25[th] and 75[th] percentiles, bar within the box represents the median, and whiskers represent the minimum and maximum value within 1.5 times of interquartile range. The dashed red line indicates zero.

2014; Jovanovic *et al*, 2015; Liu *et al*, 2017a). In a steady state within a cell system, the stable protein concentration is the result of the balance between protein synthesis and degradation (Dephoure *et al*, 2014; Battle *et al*, 2015; Liu *et al*, 2016). In a relative comparison between steady states, it is conceivable that the mRNA fold change represents the transcriptional regulation attempt, whereas the protein degradation is suggestive of post-translational adjustment attempt (Liu *et al*, 2017a, 2019; Martin-Perez & Villen, 2017). We therefore focused on unraveling the complexity of the correlation between protein degradation and mRNA abundances in the present study.

A single gene template can give rise to dozens of protein isoforms, through, e.g., the translation of mRNA AS isoforms. It was speculated that protein isoforms might exhibit drastically different stabilities that might influence their function and regulation models (Hinkson & Elias, 2011; Zecha *et al*, 2018). In the light of this, we provided correlation analysis with an AS isoform resolution. The usage of HeLa cells partially eliminates confounding factors like SNPs and individual genomic mutations, which are known to affect protein stability (Wang & Moult, 2001; Reumers *et al*, 2005; Greig *et al*, 2015; Milenkovic *et al*, 2018). Because proteins carry out virtually most cellular functions, future analysis will be extremely useful to underpin the cellular role of AS isoforms, by answering how many and what kind of alternative transcripts are translated and stabilized in the cell (Floor & Doudna, 2016; Weatheritt *et al*, 2016).

The current level of sensitivity, reproducibility, and working principle of the widely used bottom-up MS techniques (Aebersold & Mann, 2016) seems to be less efficient in analyzing the protein AS products (Smith & Kelleher, 2013; Abascal *et al*, 2015; Tress *et al*, 2017b; Wang *et al*, 2018b; Chaudhary *et al*, 2019). Previous large-scale proteomic experiments seemed to identify only a short list of AS forms, which are often low-abundant, temporal, conserved, and subtle in functional annotation (Zhu *et al*, 2014; Tress *et al*, 2017a). In this study, *firstly,* we applied a peptide-centric analysis of DIA-MS, which provides a similar quantitative consistency as SRM, but in an analytical scale far beyond the detection of hundreds of peptides (Gillet *et al*, 2012; Rost *et al*, 2015; Blencowe, 2017; Bruderer *et al*, 2017; Tress *et al*, 2017b; Amon *et al*, 2019; Mehnert *et al*, 2019) and provides a decent sensitivity (i.e., 2.5 times more peptides detected when compared to the previous report). *Secondly,* we used an mRNA abundance directed approach to identify and quantify signals of isoform groups. However, even with the most reproducible MS techniques such as DIA-MS, the biggest obstacle of analyzing AS isoform diversity by proteomics is still the sensitivity limit. For example, if we use the ultimate stringency and only accept unique proteotypic peptides which can always distinguish single distinctive isoform (i.e., no isoform group is allowed), we can only map a total of 30 genes with multiple AS events by DIA data. The transcript abundance-oriented detection of AS peptides enables the detection of exons only with appreciable read counts (Lau *et al*, 2019). In addition, the transcript abundance-oriented quantification of AS enables a wider functional annotation.

Importantly, in this study, we have implemented the full advance of DIA-MS in labeling proteomics such as pSILAC analysis. We have shown previously that the high reproducibility of DIA-MS (exemplified by SWATH-MS) boosted the efficiency of pSILAC experiment, because the pulse-chase time course routinely involves multiple samples (Rost *et al*, 2016; Liu *et al*, 2017a, 2019). Herein, to further establish pSILAC-DIA approach, we have developed the workflow, which features in (i) improved detectability of heavy isotopic signals

in the early pulse-chase time points by ISW; (ii) automatic generation of missing labeling channels from both DIA and DDA data; (iii) aligned H/L elution without the need of post-identification feature alignment. We showed the direct pSILAC-DIA approach offers especially better quantitative accuracy and richer quantitative features than the alternative methods such as pSILAC-MS1 (Schwanhausser *et al*, 2011; Mathieson *et al*, 2018) or pSILAC-TMT analysis (McAlister *et al*, 2014; Welle *et al*, 2016; Savitski *et al*, 2018; Zecha *et al*, 2018) as well as remarkable flexibility and potential. We expect our approach will be widely used in future pSILAC studies. More broadly, we suggest that there will be a pressing need to increase the multiplexity of DIA measurement in the near future (Wu *et al*, 2014; Di *et al*, 2017; Liu *et al*, 2017a, 2019). The exemplified ISW and the labeled workflows will be useful to meet this need.

We dissected the biological correlation between mRNA and $k_{loss}$ from two angles: the absolute and the relative (Liu *et al*, 2016). We found that the absolute correlation is heavily dependent on the concentrations of mRNA and protein. For example, the heterogeneity of basic degradation rates in mitochondrial ribosome and matrix can be largely explained by the rank of their gene expression levels. By profiling the relative mRNA–$k_{loss}$ correlation, we have discovered GOBP and GOCCs with substantial protein degradation difference between HeLa CCL2 and Kyoto cells. Importantly, we expect that many genes involved in these processes can serve as a list to predict the "preferably regulated" turnover events, especially when the up- or down-regulations of mRNA are determined between different steady states. This is because of the conservation of molecular basis of protein degradation, such as the involvement in stable protein complexity, chaperon networks, and ubiquitin-dependent pathways (Cambridge *et al*, 2011; Bhattacharyya *et al*, 2014; Liu *et al*, 2017a; Martin-Perez & Villen, 2017).

Our analysis uncovered two examples showing the protein degradation diversity at the sub-organelle scale: the buffering degradation of 20S proteasome subunits (but not 19S) and the buffering effect on mitoribosome large subunits (but not the small subunit). Notably, these significant proteostasis differences are derived from relative correlation analysis and are unlikely to be tightly dependent on gene expression levels. Several reports have indicated differential stability of the two proteasome subunits (Mathieson *et al*, 2018). Becher *et al* (2018) used a thermal proteome profiling strategy by MS and reported that during the cell cycle the proteasome showed a different stability and abundance variation in a protein subset of the 19S regulatory sub-complex. Using a similar approach, Volkening *et al* (2019) reported that the regulatory 19S complex has much lower protein melting temperature and a higher degree of conformation flexibility. The stability of 19S and 20S subunits can be tightly linked to their functions: The protein 20S core subunit is highly structured, whereas the 19 regulatory base and lid are composed of proteins with diverse functions in, e.g., ubiquitin recognition and protein transporting (Volkening *et al*, 2019). Consistently, our result indicates the 20S variation across cancer cells is significantly buffered by protein degradation, reinforcing the results from thermal stability profiling. The mitoribosome performs protein synthesis inside mitochondria (Greber & Ban, 2016). Our data suggested a lack of protein degradation control for the mitoribosome small subunit, which has lower mRNA–$k_{loss}$ correlation, much higher mRNA–protein correlation, and ultimately higher protein abundance variation when compared the large subunit. Similar to proteasome, this might suggest that the large subunit mitoribosome proteins have to be tightly controlled at the protein level for a potentially rate-limiting

function. To mechanistically define the relationship between protein degradation difference and sub-organelle's biological function, following environmental variance will be ultimately interesting (Romanov *et al*, 2019), but goes beyond the scope of current study.

Differential expression levels and degradation rates were observed for individual proteoforms of the same gene, according to both absolute and relative comparisons. Consistent to previous reports (Abascal *et al*, 2015), many of them are highly expressed or conserved splice variants, such as ATPase Na$^+$/K$^+$-transporting subunit alpha 1 (ATP1A1), SNW domain containing 1 (SNW1), tropomyosin alpha-1 chain (TPM1), and alpha-4 chain (TPM4). The TPM1 AS events were discovered to be informative in prognostic predictors for head and neck cancer (Liang *et al*, 2019), dilated cardiomyopathy (Pugh *et al*, 2014; Abascal *et al*, 2015), and migration of esophageal cancer cells (Huang *et al*, 2017), demonstrating the functional diversity of TPM1 AS isoforms. Unlike previous studies, we further quantified significant $k_{loss}$ difference in both basic levels and relative CCL2-versus-Kyoto ratios between TPM1 AS isoform groups. Thus, the TPM1 isoform expression stability might be associated to the phenotypic variability of texture contrast of actin structures between HeLa cell lines (Liu *et al*, 2019).

As for the significant AS switch events, we found that the loss of intron retention effectively increased the corresponding protein expression along the long-term cell passaging. Because RI may trigger NMD of mRNA or induce translational inhibition, it has been considered as an abnormal AS event associated with various diseases or stress response (Wong *et al*, 2016; Morgan *et al*, 2019; Parenteau *et al*, 2019). Our results did not find significant protein degradation regulations for RI-to-coding isoforms, which might suggest that the major effect of gene RI happens during transcription or translational levels. The clear role of proteostasis in RI may depend on the disease types (Adusumalli *et al*, 2019) and remains to be explored.

In conclusion, we applied an integrative proteomic approach and quantified the isoform-specific post-transcriptional and post-translational control across human cancer cell lines. The mRNA level, protein abundances, and protein degradation rates are all resolved to isoform levels, providing a better resolution than the gene-centric analysis. Especially, we uncovered diversity of protein turnover between different biological processes, organelles, subunits of organelles, and individual isoforms. The data argue for the necessity of more systems biological studies in the future to study the mRNA–$k_{loss}$ relationships across variable conditions such as healthy and disease states.

## Materials and Methods

### Reagents and Tools table

| Reagent/resource | Reference or source | Identifier or catalog number |
|---|---|---|
| **Experimental models** | | |
| HeLa cell lines panel | Liu *et al* (2019), NBT | |
| **Cell culture and pulse SILAC labelling** | | |
| Dulbecco's modified Eagle medium | Gibco | 41965-039 |
| SILAC Dulbecco's modified Eagle medium High Glucose medium | GE Healthcare | |
| Dialyzed FBS | PAN Biotech | P30-2101 |
| Penicillin/streptomycin | Gibco | 15140122 |
| L-lysine (13C6 15N2) | Chemie Brunschwig AG | |
| L-arginine (13C6 15N4) | Chemie Brunschwig AG | |
| L-lysine | Sigma Aldrich | L5501 |
| L-arginine | Sigma Aldrich | A5006 |
| L-proline | Sigma Aldrich | 81709 |
| **Chemicals, enzymes and other reagents** | | |
| Tris-(2-carboxyethyl)-phosphine | Sigma Aldrich | C4706 |
| Iodoacetamide | Sigma Aldrich | I1149 |
| Ammonium bicarbonate | Sigma Aldrich | 9830 |
| Sequencing-grade porcine trypsin | Promega | V5113 |
| **Software** | | |
| Spectronaut™ Professional+ | Biognosys AG | v13 |
| Perseus | Tyanova *et al* (2016) | v1.6.2.2 |
| R | R Core Team (2018) | v3.2.5 |
| GraphPad Prism | GraphPad Software, Inc. | v5.04 |
| **Other** | | |
| Orbitrap Fusion Lumos Tribrid mass spectrometer | Thermo Scientific | |
| EASY–nLC 1200 systems | Thermo Scientific | |

## Methods and Protocols

### RNA-Seq data set processing

The RNA-Seq data set used in this study was published previously (Liu *et al*, 2019). RNA-Seq data are available on GEO (GSE111485). A matrix of all uniquely mapped transcripts with their abundance in FPKM (Fragments Per Kilobase Million) in different HeLa cell variants was used as an input data set for this study. To create a sample-specific sequence library, we used the RNA-Seq data measured for all HeLa cell lines. Only those transcript sequences corresponding to protein-coding transcripts (defined by Ensembl biotype) and expressed above a conservative FPKM > 1 threshold in at least three HeLa cells variants were translated into corresponding protein sequences. For testing purpose, the different cutoffs with FPKM > 0.1 and FPKM > 0 were also used for a smaller data sets (five raw files of HeLa 1 were used for library generation; three pSILAC-DIA raw files were used for targeted data extraction).

### HeLa cell lines and sample processing

The MS samples used in the present study are from the samples prepared for the previous study where a detailed description of all experimental procedures was already provided, including HeLa cells collection from different laboratories, central cultivation, pSILAC experiment design and procedure, protein extraction, and sample preparation for LC-MS analysis (Liu *et al*, 2019). Briefly, a uniform protocol following a routine cell culture guideline was used at each site to prepare the cells for shipment to the central laboratory, cultured in 5% $CO_2$, 37°C, Dulbecco's modified Eagle's medium (Gibco). Both total proteomic and pulse-chase SILAC proteomic samples were prepared to measure protein abundance and degradation rate (Liu *et al*, 2019). From the originally analyzed fourteen HeLa cell lines, two cell lines were excluded from the re-analysis presented in this report. First, the HeLa 11 was excluded because of its deviating genome dosage type (Liu *et al*, 2019). Second, the HeLa 5 was excluded as it represents a HeLa S3 (CCL2.2) cell line. The remaining twelve cell lines represented six HeLa cells variants subtype CCL2 (2, 6, 7, 12, 13, and 14) and six HeLa cells variants subtype Kyoto (1, 3, 4, 8, 9, and 10), presenting a balanced comparison between HeLa Kyoto and CCL2 strains. The pSILAC experiment (Liu *et al*, 2019) was performed as follows:

1   SILAC DMEM High Glucose medium (GE Healthcare) lacking L-arginine and L-lysine was first supplemented with light or heavy isotopically labeled lysine and arginine, 10% dialyzed FBS (PAN Biotech), and 1% penicillin/streptomycin mix (Gibco). Specifically, 146 mg/l of heavy L-lysine (13C6 15N2) and 84 mg/l of L-arginine (13C6 15N4) (Chemie Brunschwig AG) and the same amount of corresponding unlabeled amino acids (Sigma-Aldrich) were supplemented, respectively, to configure heavy and light SILAC medium. Additionally, 400 mg/l L-proline (Sigma-Aldrich) was also added to SILAC medium to prevent potential arginine-to-proline conversion.

2   HeLa variants were first cultured on 15-cm cell culture dishes in pre-prepared light SILAC medium and stabilized in culture for 3–4 days.

3   Upon release of cells by 0.25% trypsin/EDTA, cells were counted using a Neubauer hemocytometer.

4   Subsequently, six 10-cm dishes were prepared for each cell variant with a seeding density of $1.5 \times 10^6$ cells per plate, corresponding to three time points with two replicates each.

5   The cell culture plates were incubated for 14 h, at 5% $CO_2$ and 37°C, overnight.

6   Cells were washed three times with PBS at 37°C.

7   The medium was replaced by heavy SILAC (K8R10) medium.

8   Cells were harvested and counted in two biological replicates at four different time points (0, 1, 4.5, and 11 h). Two dishes of whole-process replicate were prepared at each time.

9   The cell pellets were snap frozen in liquid nitrogen after removal of the PBS and stored at −80°C.

### Data acquisition on Orbitrap Lumos mass spectrometer

For this study, the Orbitrap Fusion Lumos Tribrid mass spectrometer (Thermo Scientific) coupled to a nano-electrospray ion source (NanoFlex, Thermo Scientific) was used as the liquid chromatography-mass spectrometry (LC-MS) system for performing both data-dependent acquisition (DDA) and data-independent acquisition (DIA), as previously described (Li *et al*, 2019; Mehnert *et al*, 2019). Peptide separation was carried out on EASY-nLC 1200 systems (Thermo Scientific, San Jose, CA) using a self-packed analytical PicoFrit column (New Objective, Woburn, MA, USA; 75 μm × 25 cm length) using C18 material of ReproSil-Pur 120A C18-Q 1.9 μm (Dr. Maisch GmbH, Ammerbuch, Germany). Buffer A was composed of 0.1% formic acid in water, and buffer B was composed of 80% acetonitrile containing 0.1% formic acid. To separate the HeLa peptide mixtures for both DDA and DIA measurements, a 2-h gradient with buffer B from 5 to 37% at a flow rate of 300 nl/min was conducted.

For DDA-based proteomics, the MS1 scan range setting was from 350 to 1,650 $m/z$ with the RF lens 40% (Li *et al*, 2019). The MS1 resolution was kept at 120,000 at $m/z$ 200. The AGC value was 5.5E5, and the maximum injection time was 40 ms for MS1. For MS2, the top speed (cycle time 3 s) was used, meaning that the numbers of data-dependent scans were maximized in each cycle time if the desired resolution and AGC were achieved. HCD collision energy was 28%. The dynamic exclusion parameters were set to ensure that the already sequenced precursors were excluded once from reselection for 30 s. The isolation window was 1.2 $m/z$, and the MS2 resolution was 15,000. The AGC value and the maximum MS2 injection time were set to 5eE4 and 35 ms, respectively. All the data were collected with the 2-h gradient LC method as described above.

For DIA-based proteomics, our DIA-MS method was configured to consist one MS1 survey scan and 40 MS2 scans of variable windows (Li *et al*, 2019; Mehnert *et al*, 2019). The MS1 scan range is 350–1,650 $m/z$, and the MS1 resolution is 120,000 at $m/z$ 200. The MS1 full-scan AGC target value was set to be 2.0E5, and the maximum injection time was 100 ms. The MS2 resolution was set to 30,000 at $m/z$ 200, and normalized HCD collision energy was 28%. The MS2 AGC was set to be 5.0E5, and the maximum injection time was 50 ms. The default peptide charge state was set to 2. Both of MS1 and MS2 spectra were recorded in profile mode.

### Targeted data extraction of protein expression data

All DIA-MS data analyses were performed using a specific version of Spectronaut™ Professional+ (all the features are now available in Spectronaut v13).

1  *Library generation*: A hybrid assay library was generated including all protein expression samples measured in technical triplicates using both DDA and DIA-MS method (84 raw files). Furthermore, we included raw files that were generated from fractionated HeLa samples to make an ultra-comprehensive spectral library (provided with the raw mass spectrometry datasets) for our single-shot DIA measurements (122 raw files in total, 163,333 precursors corresponding to 11,847 proteins in total). For the library generation, we applied the default BGS Factory Settings for both Pulsar Search and Library Generation and used the FASTA protein sequence database we generated as described above.

2  *Targeted data extraction*: The targeted data extraction was performed using the default BGS Factory Settings. Briefly, full tryptic digestion allowing two missed cleavages, carbamidomethylation as a fixed modification on all cysteines, oxidation of methionines, and protein N-terminal acetylation as dynamic modifications were set. Both precursor and protein FDR were controlled at 1%. For quantification, "Qvalue" workflow was used for filtering; mean precursor quantity was used for peptide quantification (TOP 3); peptides were grouped based on stripped sequences. Cross Run Normalization was performed using "Global Normalization" on "Median" normalization strategy. Interference correction was enabled with min 3 MS2 precursors to keep. Other parameters were kept as default unless specified.

### Targeted data extraction of pulsed SILAC data

For pSILAC data analysis, we generated an additional SILAC hybrid library.

1  *Library generation*: We used the FASTA protein sequence database and search archives from the label-free (i.e., "light") library generation as described above and further combined these with pSILAC-DIA-MS raw files (see also Appendix). For the pSILAC-DIA data extraction, the default BGS Factory Settings for Pulsar search was used with modification in the Labeling setting. "Labelling Applied" option was enabled, and SILAC labels ("Arg10" and "Lys8") were specified in the second channel. For the library generation step, the default BGS Factory settings for library generation was used with two important modifications. First, a complete labeling of the whole library was ensured by selecting the "*In-Silico* Generate Missing Channels" option in the Workflow settings. Secondly, we completely excluded b-ions from the library in the Spectral Library Filters for the analysis of the data presented in the study to avoid potential interference and ratio distortion caused by b-ions not carrying the heavy labels. In total, 164 raw files were used for the pSILAC library generation, and the final library contained 159,963 peptide precursors corresponding to 11,623 proteins. All precursors in the library were present in both light and heavy versions with TOP 6 corresponding light and heavy fragments enabled for targeted data analysis. The retention time drift is handled by using iRT space and a method for high-precision prediction of RT in targeted DIA analysis in Spectronaut, as previously described (Bruderer *et al*, 2015, 2016).

2  *Targeted data extraction*: To perform pSILAC data identification and quantification, the Inverted Spike-In workflow was used. In the Workflow settings, the "Spike-In" workflow was selected in Multi-Channel Workflow Definition. Importantly, both "Inverted" and "Reference-based Identification" options were enabled. The hybrid library generation and analysis of pSILAC data together with the main advantages of the Inverted Spike-In workflow are in detail described in the Appendix.

We also note here that the pSILAC workflow was further optimized, and now, the use of b-ions is possible in Spectronaut v13 which further increases the number of identified and quantified peptides by 14% (and protein identifications by 11.2%). These important optimizations are also described in the Appendix.

### Protein degradation rate estimation by pSILAC—"RIA" method

pSILAC enables protein degradation rate calculation by monitoring the intensities of light and heavy peptides across several time points and using this information to fit a model for protein degradation rate estimation (Pratt *et al*, 2002; Doherty *et al*, 2009; Claydon & Beynon, 2012; Rost *et al*, 2016).

In the experiments using pSILAC, the working assumption is that the cells are, respectively, maintained growing in a steady state (i.e., without perturbation), so that for a given protein with known concentration, the degraded and synthesized protein copies are balanced. Under this assumption (Claydon & Beynon, 2012), the respective determination of the protein-specific $k_{loss}$ within each HeLa cell line directly estimates protein turnover behavior in that cell line. To perform the estimation, we used a similar approach as was employed in our previous studies (Rost *et al*, 2016; Liu *et al*, 2017a, 2019).

1  At each time point, the amount of heavy (H) and light (L) precursor was extracted and used to calculate the relative isotopic abundance $RIA_t$.

$$RIA_t = \frac{L}{L + H}$$

2  This is analogous to Pratt *et al* (2002) and others. The value of $RIA_t$ is time dependent, as unlabeled proteins are replaced with heavy-labeled proteins during the course of the experiment. This is due to dilution of the cells as well as intracellular protein turnover, where the rate of loss can be modeled as an exponential decay process.

$$RIA_t = RIA_0 \cdot e^{(-k_{loss} \cdot t)}$$

where $RIA_0$ denotes the initial isotopic ratio and $k_{loss}$ the rate of (hourly) loss of unlabeled protein. We assumed $RIA_0 = 1$, as no heavy isotope was present at $t = 0$, thus the value of $RIA_t$ will decay exponentially from 1 to 0 after infinite time and used nonlinear least-squares estimation to perform the fit. As discussed in Pratt *et al* (2002), these assumptions may reduce measurement error especially at the beginning of the experiment, where isotopic ratios are less accurate owing to the low absolute number of heavy precursor ions where our new pSILAC-DIA strategy is helpful.

3   A weighted average of the peptide precursor $k_{loss}$ values was performed to calculate the $k_{loss}$ values for all unique peptide sequences. We excluded precursors quantified in a single time point only and only included peptides with increasing isotope ratio over time.

4   Only those peptide $k_{loss}$ values assayed in every cell sample were accepted for cross-comparison.

5   The $k_{loss}$ for each protein isoform groups was computed as the median of all peptide-level rates.

6   In proliferating cells, this parameter has two components, the degradation rate ($k_{deg}$) and dilution of the protein pool by exponential growth of the cell culture, which is described using the cell division rate ($k_{cd}$):

$$k_{loss} = k_{deg} + k_{cd}$$

Thus, to estimate protein "degradation rates", the cell division rates are subtracted from the $k_{loss}$ rates:

$$k_{deg} = k_{loss} - k_{cd}$$

7   However, as previously documented (Liu *et al*, 2019), to avoid the possible calculation issues due to light amino acid recycling (Boisvert *et al*, 2012), inaccurate cell doubling time measurement in different HeLa cells, and the fact the used cells are all HeLa strains, we use $\log_2 k_{loss}$ [$k_{loss}$ as a short name in the text; (Claydon & Beynon, 2012)] as a proxy estimate to protein degradation rate (Liu *et al*, 2019), whenever applicable, to perform the cross-cell and multi-omics comparisons and visualizations.

### Protein degradation rate estimation by pSILAC—"NLI" method

High accuracy of our pSILAC-DIA measurement enables a direct analysis of L and H peptide intensities separately. In addition to the RIA method, it is possible to determine a *de facto* protein degradation rate directly from the rates of loss from the L (unlabeled) intensities. To perform this calculation termed normalized light intensity (NLI)-based method, firstly, we normalized the DIA measurements based on the sum of total heavy and light signals across time. Then, we extracted the light-channel quantities and estimated the degradation rate by fitting the desired decay curve on each peptide precursor using the same algorithm as described above in the RIA method description. To compare this approach to RIA, the data were further processed in the same way to enable a direct comparison.

### Sample-specific peptide assignment to protein AS groups

1   Peptide intensities (defined as unique peptide stripped sequences) were exported from Spectronaut. As for protein identities (IDs), all IDs mapping to a peptide were preserved to keep full information about the splicing isoforms that map to a peptide.

2   The data were $\log_2$ transformed, triplicate measurements were averaged for all HeLa cell lines (median value; the peptide had to be quantified in at least two injections), and only peptide full profiles (i.e., peptides quantified in all twelve HeLa cell lines) were kept for the downstream analysis.

3   Peptides mapped to several genes were excluded and were further classified based on the criteria whether they map uniquely to one splicing isoform of a gene (*unique* peptides) or

to multiple splicing isoforms of the same gene (*shared* peptides).

4   Both unique and shared peptides were then collapsed to create a matrix of quantified proteins and protein groups, respectively (by summing; at least two peptides were required for a protein/protein group).

5   Similar approach was applied for the pSILAC data. After $k_{loss}$ calculation, only peptide $k_{loss}$ full profiles were used with peptides mapping to multiple genes removed.

6   The $k_{loss}$ values were $\log_2$ transformed and collapsed to create a protein and protein group matrix (average; at least two peptides were required for a protein/protein group).

7   The two data sets were then merged in a protein-centric way; i.e., the protein $k_{loss}$ values were mapped to the protein expression matrix using the protein/protein group IDs.

8   To map the mRNA abundance data to the assembled protein matrix, we exploited the fact that in our protein FASTA DB, each entry was annotated by a unique Ensemble transcript ID (ENST) and thus could be easily mapped to its corresponding transcript abundance in the RNA-Seq data (i.e., FPKM).

9   We first mapped the unique proteins (UQ) with the abundance of the corresponding transcripts.

10  For the protein groups quantified based on shared peptides quantities, we applied a sample-specific protein inference similar to a strategy described previously (Liu *et al*, 2017b). For each splicing isoform included in a shared protein group, we retrieved its average abundance on mRNA level (the average was calculated from all HeLa variants).

11  The major (i.e., the most abundant) splicing isoform on mRNA level was then selected as the best representative transcript ID for the whole protein group (*shared major, SM)* and was used for transcript abundance mapping.

### Identification of alternative splicing switch events

Alternative splicing switch events were identified using SwitchSeq (preprint: Gonzàlez-Porta & Brazma, 2014).

1   The RNA-Seq data for HeLa 12 (variant CCL2, passage 50; P50) were compared to the data from HeLa 14 (variant CCL2, passage 7; P7). Both cell lines were analyzed in three biological replicates.

2   The input matrix for SwitchSeq was pre-filtered to only contain matrix with non-zero values in all six samples and FPKM > 1 in at least three samples.

3   Ensembl v87 was used to retrieve transcript biotypes; gene expression threshold of 1, expression breadth of 50, and dominance of 1 were used as parameters for SwitchSeq analysis (preprint: Gonzàlez-Porta & Brazma, 2014).

### Calculation of the genome-wide, protein AS group-specific correlation across HeLa cell lines

For each protein AS group, 12 data points ($x$, $y$), each one of them representing the mRNA abundance ($x$) and protein $k_{loss}$ ($y$) in one of the twelve cell lines, were used to calculate a protein AS group-specific Spearman's *rho* value to describe the mRNA–$k_{loss}$ correlation. This value was calculated for each one of the protein AS isoform group in the data set. To calculate

the protein AS group-specific correlation between other layers (i.e., mRNA–protein and protein-$k_{loss}$), the same principle was applied.

### Other bioinformatic analyses

All downstream analyses and data visualizations were performed in Perseus v1.6.2.2 (Tyanova *et al*, 2016) and in R software (v3.2.5) (R Core Team, 2018). Differential expression analysis of the RNA-Seq expression profiles between HeLa CCL2 and HeLa Kyoto was performed using R package "edgeR" (Robinson *et al*, 2010). Differential protein expression and degradation was determined using a two-sided *t*-test in Perseus (Tyanova *et al*, 2016) followed by Benjamini–Hochberg correction on the *P*-values to control the false discovery rate (FDR). In all tests, the Benjamini–Hochberg FDR was lower than 0.01 for values reported as significant. The DAVID bioinformatics resource v6.8 (https://david.ncifcrf.gov) was used to extract the protein annotations (DAVID GOCC DIRECT and DAVID GOBP DIRECT) (Huang *et al*, 2008). All boxplots, violin plots, and bubble plots were generated using R package "ggplot2". In the boxplots, the bold line indicates median value; box borders represent the 25th and 75th percentiles, and whiskers and gray panel represent the minimum and maximum value within 1.5 times of interquartile range. Outliers out of this range are depicted using solid dots. In the violin plots, the boxplots are combined with kernel density as the violin curve to show the distribution of the data. The white dot marks the median. In the bubble plots (Fig 5E and F), the size of the bubble corresponds to the number of proteins comprised in the category; the color code corresponds to *P*-value calculated by 2D enrichment analysis in Perseus. The colored scatterplots from blue-to-yellow in Fig 3 were visualized using R "LSD" package; the heatmap was created using R package "heatmap.2". The grouped scatter plots in Fig 5 were created in GraphPad Prism® v5. Prism or R were used to calculate all Mann–Whitney–Wilcoxon test *P*-values, Kruskal–Wallis test *P*-values, and pairwise Wilcoxon test *P*-values with Bonferroni correction reported in the figures. Spearman's correlation coefficients were calculated using R (functions cor() or cor.test()). 1D and 2D enrichment analyses were performed in Perseus using relative enrichment for UniProtKB protein IDs and *P*-value or Benjamini–Hochberg adjusted *P*-value thresholds as indicated (Cox & Mann, 2012). Differential splicing isoform expression and degradation analysis shown in Fig 4 was performed in R. For genes with two splicing isoforms detected, *t*-test was used to calculate if the difference between the isoforms is significant based on all peptide-level data for these isoforms. For genes with multiple splicing isoforms detected, ANOVA was used to calculate the *P*-values followed by pairwise comparison by Tukey honest significant differences test. An additional fold change cutoff was performed requiring $log_2$ fold change of at least 0.32 (for at least one pairwise comparison in the case of the ANOVA results).

## Data availability

The new mass spectrometry data from this publication (126 raw files generated by Orbitrap Fusion Lumos platform), spectral libraries, FASTA DB, and Spectronaut search results have been deposited to the following database: ProteomeXchange Consortium via the PRIDE (Perez-Riverol *et al*, 2019) PXD014847 (http://proteomecentral.proteomexchange.org).

Expanded View for this article is available online.

## Acknowledgements

This research was supported in part by Pilot Grants (to Y.L.) from Yale Cancer Systems Biology Symposium and Yale Cancer Center. B.S. was supported by grant of the Czech Academy of Sciences (L200521953).

## Author contributions

BS performed the major data analysis. HZhu, WL, CW, and HZhou acquired the Orbitrap DIA data. TG and LR contributed to the software development facilitating SILAC-DIA analysis. MF analyzed the RNA-Seq data and generated protein FASTA databases. GR, P-LG, and ZH provided critical feedback to the manuscript. YL supervised the study. BS and YL wrote the paper.

## Conflict of interest

T.G. and L.R. are employees of Biognosys AG. Spectronaut is a trademark of Biognosys AG.

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
