## [Review Process File · Molecular Systems Biology]

Isoform-resolved Correlation Between mRNA Abundance Regulation and Protein Level Degradation

Barbora Salovska, Hongwen Zhu, Tejas Gandhi, Max Frank, Wenxue Li, George Rosenberger, Chongde Wu, Pierre-Luc Germain, Hu Zhou, Zdenek Hodny, Lukas Reiter, and Yansheng Liu

Review timeline:

Submission date:	11 th August 2019
Editorial Decision:	1 st October 2019
Revision received:	11 th December 2019
Editorial Decision:	30 th January 2020
Revision received:	6 th February 2020
Accepted:	12 February 2020

Editor: Maria Polychronidou

Transaction Report:

1st Editorial Decision

1st October 2019

Thank you again for submitting your work to Molecular Systems Biology. We have now heard back from the two referees who agreed to evaluate your study. As you will see below, the reviewers acknowledge that the proposed approach seems relevant for the community. They raise however a series of concerns, which we would ask you to address in a major revision.

Both reviewers provide constructive suggestions on how to improve the study. Some of the more fundamental concerns are the following:

- Reviewer #1 refers to the need to include comparisons to other methods to better support the superiority of the proposed approach.
- Reviewer #2 requests several clarifications and suggests some additional analyses to strengthen the main conclusions.
- As reviewer #1 points out, the novelty of the work needs to be clarified and the findings contextualized compared to previous studies.

Please feel free to contact me in case you would like to discuss in further detail any of the issues raised by the reviewers.

REFeree REPORTS

Reviewer #1:

The manuscript by Salovska et al builds on an earlier report by the authors (Liu et al, Nature

Biotechnology 2019). That study reported the measurement of mRNA, protein and protein loss (turnover) rates across a series of HeLa cell lines. The current manuscript by Salovska et al reports an improved version of their original pSILAC-DIA proteomics workflow, which they use to re-analyse the samples of the NBT paper. In addition, the authors report a proteogenomic analysis of their new proteomics data, by searching them against the relevant expressed mRNA splice forms inferred from the RNA-seq data of the original report. Finally, they extend and deepen the analysis of the mRNA, protein and k-loss comparison. The key aspects of this comparison (e.g. the finding that protein degradation mainly buffers but not amplifies mRNA changes) have already been reported in the NBT paper, and isoform-resolved protein turnover rates have also been reported before (although without comparison to mRNA abundances; Zecha et al, MCP, 2018). Therefore, the development / improvement of the pSILAC-DIA workflow to determine turnover rates is the key new contribution of this paper.

Indeed, the new pSILAC-DIA workflow appears to be a quite an improvement over the previous SWATH-MS approach used in the NBT paper, which is very interesting and could be a very valuable tool for the community.

My main point of criticism is that at the moment, without reading the NBT paper, it isn't quite clear which insights are specific and novel for this paper. For example, Figs. 2D-G are almost the same as Fig. 4D-E in the NBT paper, the only difference being that here the resolution is at protein isoform level. However, this is not immediately clear when reading the paper. Moreover, it is not clear if the isoform-resolution actually makes a difference, i.e. there is no direct comparison of the two approaches, and the conclusions appear to remain the same.

Given that the method development is the key novelty, my second major point is that the authors should compare their method in more detail to other turnover methods that have recently been developed. In particular, I'm thinking about the TMT-SILAC approach developed by Welle et al (MCP, 2016), Zecha et al (MCP, 2018) and Savitski et al (Cell, 2018). What would be the advantage of using the pSILAC-DIA approach over these?

In addition, I have a few minor points, as follows:

- Please explain the use of the term "loss rate" rather than "degradation rate", and please document the pulse-labelling better. I understand the samples had been described before, but since it is a paper about loss / turnover rates it would be good to know what the labelling time points were etc...
- Differential turnover of protein isoforms: How can you be sure to have different turnover rates for different protein isoforms? Presumably most of these measurements compare single peptides? If so, how do you know that's not just a difference between the peptides of the same protein? In general in these assays there appears to be quite a range of turnover rates identified by different peptides for the same protein (e.g. see Welle et al)
- The title gives the impression that both mRNA and protein turnover were measured
- Page 4, line 9: I think you mean "sequence" database not "proteomic" database?
- 6,552 proteins using a single-shot analysis seems amazing. For how many proteins do you actually have loss rates and in how many samples? How many missing values are there? Essentially, please document the dataset a little better
- Fig. 2D-G: No axis labels
- The discussion is overstating the claims of novelty of the study quite considerably, e.g. the first (page 17, line 11) and second (line 13) points are not specific for this paper.

Reviewer #2:

Salovska et al. addressed the question of how protein levels, mRNA levels and protein turnover relate to each other. A single gene can give rise to many transcript isoforms, which then are

translated into different proteoforms, very often with unique functions within the cell. In most previous integrated proteomics studies the regulation of gene expression of different proteoforms was not directly assessed. Salovska et al. generate an impressive data set that indeed allows the study of the interplay between mRNA levels, protein levels and protein with a special focus of their analysis to a quite large panel to alternative spliced isoforms, which provides significant novelty. Moreover, the peptide centric analysis by DIA of pulsed SILAC (pSILAC) data is a very nice new way to analyze pSILAC data and could be extremely useful for the community. They applied their novel approach to gain new insight about gene expression across about a dozen different strains of HeLa cell lines and reported a variety of different regulatory patterns between cells and for different gene groups. All in all I think the authors generated a very impressive and intriguing data set that definitely would be of great use for the gene expression community and could potentially also serve as a blueprint for studying different layers of gene expression. However, I think before acceptance of the manuscript can be considered several issues absolutely need to be addressed.

The main issues are the following:

1. It is hard to understand if the authors mean protein degradation, protein loss or protein turnover. Although the first two terms the same, the third is not really the same as this is affected by protein production and protein loss, but this is actually what I think they are measuring and should be using. However, the authors constantly push the term Kloss, which would indicate that they only look at protein degradation and therefore should be only dependent on the light signal. Yet, if one reads through materials and methods it seems that the authors calculate Kloss as $L/(H+L)$, but that is not really the loss of protein, but the percentage of L of the total protein and depends on H and therefore protein production. Again true protein degradation should only look at the L signal over time (and use the H+L signal just for normalization for same sample input). The way the authors calculate Kloss profoundly affects the interpretation of the results. For example the loss of L (protein degradation) could be the identical between the different cell types, but the protein production (there for the H signal)could be different between the cell types (e.g. mRNA is upregulated and more protein is produced), but the way the authors calculate protein degradation and just by H being different it would look like a higher Kloss for the authors, although it is just higher turnover not higher loss. It should be noted that in such a case as described there should also be a higher amount in total protein. So in reality that would not be buffering but an amplification or neither or as for example if mRNA is up I would expect H to go up also, without L changing. So that would just be all driven by mRNA change, but in their formula it would look like that Kloss decreases to buffer, which it actually does not. So especially conclusion if something is amplified or buffered have to be taken with caution with the way the authors calculate Kloss.

It might easily be that I misunderstand something and I am missing something obvious, but if I got it correctly how the authors calculated Kloss then at the present state of my understanding I do not agree with many of their conclusions. This could be easily addressed by reanalysis of the data (e.g. calculating the real protein loss).

2. Data presented in Figure 4 (and also partly Figure 2): all the correlation - what has been correlated with what for each gene. For example in 4A in the mRNA to Kloss plot - did you compare the mRNA intensity value to the Kloss value for each gene and this for each cell line separately and these 12 data points gave the spearman rho for each gene? This was not completely clear to me - please clarify? This is also true for Figure 2 - not really 100% clear on what all the correlations are based on - e.g. Figure 2H to 2J. It is essential to understand how that was calculated exactly. I might have missed it, but did also not find that information in materials and methods.

More minor issues:

In general the manuscript is not an easy read and sometimes I was not sure what the authors mean.

In particular:

- Page 4, Line 15 to 17: that sentence seems to have important information but the way it is presented it is hard to get. Maybe elaborate in one or two sentences more.
- Page 6, third paragraph, Lines 15 to 27: based on the description I am not 100% sure what you did. Did you actually spike in peptides or just use the light signal already present from the pSILAC?
- Page 13, Line 28 and 29: "less stabilized by turnover" - not sure what that means - degraded faster? Turned over faster?

- Page 16, Line 15 and 16: The author write rightly that protein turnover is "denotes the balance between protein synthesis and degradation for a final product.", but are constantly talking about Kloss. What is it now?
- Page 18, Line 26 till 28: I do not really fully understand that sentence - maybe rephrase?

Other minor issues:

- Page 9, Line 9 to 11: Maybe I am missing something obvious, but I would actually suggest that the transcriptional induction leads to faster protein turnover not a surprise as more protein is produced upon induction and therefore also turnover increased.
- Page 9, Line 17 to 20: These correlations are extremely weak and I think that these kind of conclusions like that protein turnover buffers mRNA variation are extremely strong for such weak correlations, even if significant (which is not surprising with the number of data points).
- Page 10, Line 4 to 6: Again the values are very small and it is hard to believe that this is really the case at such an effect size.
- Page 12, Line 13 and 14: This is simply not true as far as I understand they are looking at turnover and this influenced by production and degradation and therefore higher production at the same rate of degradation can also lead to faster turnover and it is not necessary degradation. They should put this then at least in relation to total protein (as this should be higher in such an example).
- Page 13, Line 10 to 12: how does the total protein level look for those categories. Also went up - if so no buffering necessarily, but potentially even amplification.

Again, I want to stress that this is actually a very nice data set and I do absolutely think publication of it in MSB would be of great interest to the proteomics and gene expression community, but at the current state certain things need to be addressed first. Having said that I am positive that the authors can address most of my concerns.

1st Revision - authors' response

11th December 2019

Reviewer #1:

The manuscript by Salovska et al builds on an earlier report by the authors (Liu et al, Nature Biotechnology 2019). That study reported the measurement of mRNA, protein and protein loss (turnover) rates across a series of HeLa cell lines. The current manuscript by Salovska et al reports an improved version of their original pSILAC-DIA proteomics workflow, which they use to re-analyse the samples of the NBT paper. In addition, the authors report a proteogenomic analysis of their new proteomics data, by searching them against the relevant expressed mRNA splice forms inferred from the RNA-seq data of the original report. Finally, they extend and deepen the analysis of the mRNA, protein and k-loss comparison. The key aspects of this comparison (e.g. the finding that protein degradation mainly buffers but not amplifies mRNA changes) have already been reported in the NBT paper, and isoform-resolved protein turnover rates have also been reported before (although without comparison to mRNA abundances; Zecha et al, MCP, 2018). Therefore, the development / improvement of the pSILAC-DIA workflow to determine turnover rates is the key new contribution of this paper.

Indeed, the new pSILAC-DIA workflow appears to be a quite an improvement over the previous SWATH-MS approach used in the NBT paper, which is very interesting and could be a very valuable tool for the community.

>In addition to this reviewer's summary (and we will show that we significantly improved

the presentation of method development following this reviewer suggestion later), we would like to briefly point out:

a) One of the key aspects in the current paper is the dissection of *relative* and *absolute* correlation analysis between mRNA regulation and protein degradation dynamics. In this regard, Zecha et al did not analyze the *relative* comparison scenario in the large scale, for which we think is even more important than the *absolute* level analysis - because the relative change defines the “regulation” between samples. Indeed, distinctive biological insights were obtained from our absolute and relative analysis. We suggest that such a view is essential for the interpretation of biological correlations (Please see a relevant perspective from Vogel and colleagues, <https://www.embopress.org/doi/full/10.15252/msb.20167325>).

b) Other novelties: The relative correlation analysis in this study highlights both buffering and concerting trends through protein degradation control (instead, only buffering was discussed in the NBT paper, which was widely appreciated as a paper delivering the message that HeLa cells used in different labs are heterogeneous). We further reported the sub-organelle diversity in mRNA-*k*_{loss} correlation and the impact of intron retention on protein expression. Together, our deeper data analysis on a newly acquired, larger dataset (although with a published sample set) yielded novel biological insights that are not only incremental. We now improve our discussion accordingly to highlight these novelties (**Line 6, Page 16; Line 9-22, Page 17**).

My main point of criticism is that at the moment, without reading the NBT paper, it isn't quite clear which insights are specific and novel for this paper. For example, Figs. 2D-G are almost the same as Fig. 4D-E in the NBT paper, the only difference being that here the resolution is at protein isoform level. However, this is not immediately clear when reading the paper. Moreover, it is not clear if the isoform-resolution actually makes a difference, i.e. there is no direct comparison of the two approaches, and the conclusions appear to remain the same.

> We would like to clarify that the above-mentioned figures 2D-G were produced from unique hits (UQ) and shared major (SM) hits following the isoform perspective. Indeed, in these figures, the general trend of the isoform-specific result is largely the same to the gene-centric study (note the correlation coefficient has slightly improved). Our main intention was to show that UQ and SM plots are very similar (i.e., our analysis did not generate any deviation for SM, taking UQ is the quality control), which is an important basis for our following up study. We now add an explanation to clarify this (**Line 11, Page 8**).

The general similar correlation trends and coefficients achieved by isoform- or gene-resolution analysis respectively may be not surprising, but is still a strong global statement (this means, alterations in differential transcript usage and gene expression alter protein abundance and turnover proportionate to transcript levels). Besides the global observation, it is conceivable that our isoform-resolution analysis provides the individual protein examples whose expression and turnover vary between isoforms (now **Figure 4E-4H**), as

well as the opportunity to study the proteomic impact from isoform-switching events such as intron retention (now **Figure 6**). These are not possible with gene-centric analysis.

Following this reviewer's concern, we have further clarified these aspects in the revision (above, and also **Line 17, Page 16; Line 28 Page 15**).

Given that the method development is the key novelty, my second major point is that the authors should compare their method in more detail to other turnover methods that have recently been developed. In particular, I'm thinking about the TMT-SILAC approach developed by Welle et al (MCP, 2016), Zecha et al (MCP, 2018) and Savitski et al (Cell, 2018). What would be the advantage of using the pSILAC-DIA approach over these?

> We take this major comment seriously and thank this reviewer for raising it up. Following this suggestion, we have now significantly improved the presentation and discussion of the pSILAC-DIA methodology development and think that MSB is a good journal to deliver the advantages of pSILAC-DIA. We think this point strengthened our study.

These improvements involve one main figure (**Figure 2**), five supplementary figures (**Appendix Figures S2-S6**), and a **Supplementary Note 3** included in the **Appendix** pdf (text pasted below in the reply).

Supplementary Note 3: Comparison of pSILAC-DIA to pSILAC-MS1 and pSILAC-TMT approaches for protein turnover analysis.

Below we compared our pSILAC-DIA data to two alternative methods for the proteome-wide turnover measurement by pSILAC experiment – namely pSILAC-MS1 and pSILAC-TMT. Both methods are often used. For example, Savitski and colleagues are the major developers of the methods and software for the multiplexed TMT-based workflow analyzing protein thermal stability (Franken et al, 2015) and dynamics. In 2018, they published a systematic analysis on protein turnover in primary cells using an optimized pSILAC-MS1 workflow (Mathieson et al, 2018) and a study uncovering proteolysis targeting chimeras (PROTACs) effects using a pSILAC-TMT workflow (Savitski et al, 2018).

pSILAC-DIA vs. pSILAC-MS1:

In pSILAC-MS1 approach, the same pSILAC experiment is performed, whereas the SILAC heavy/light ratios are inferred from MS1 isotopic pairs following a shotgun proteomic workflow.

- 1) In Mathieson et al., with the extensive, gel- or high-pH-based fractionations coupled mass spectrometry quantification, 4000-6000 proteins were on average identified in each of the several primary cell types (Mathieson et al, 2018) (a further check on their supplementary table suggests ~2200 proteins were quantified with turnover estimates across cell types). This number is very close to our DIA data reported here without any fractionation before MS analysis. We suggest that the nature of DIA (or SWATH-MS in our previous papers (Liu et al, 2017; Liu et al, 2019)) generates much less missing values than traditional shotgun (or DDA) measurement, and thus favorably supports those experiments involving multiple

(or large number of) samples to be measured, such as the ones with multiple time points following pSILAC design.

- 2) In Mathieson et al, the authors had to optimize the data analysis workflow of pSILAC-MS1 by two innovations (Mathieson et al, 2018). **a)** They improved the theoretical MS1 isotope-fitting algorithm, because the MS1 ions features have multiple, (possibly) high charge-states. **b)** They improved the MS1 isotope dissection to ensure the quantification quality (especially for early time points), particularly because certain co-eluting interfering isotopic clusters are indistinguishable. Although these improvements were impressive as compared to the traditional pSILAC-MS1 experiments, they are currently only available with isobarQuant search engine and package (Mathieson et al, 2018).

In our pSILAC-DIA, correspondingly to above, **a)** With the new **Appendix Figure S2**, we have shown that MS2 based DIA quantification has much less and more uniformly distributed charge states (83.08% are charge 1, and the other 15.92% are charge 2, **Figure S2A**) than the MS1 results (**Figure S2B**), which greatly reduced the isotope-fitting difficulty for calculating theoretical isotopic envelope. **b)** Furthermore, because all high-resolution DIA-MS2 peak groups (with 30k in Orbitrap) are aligned and identified at MS2 level, the heavy and light signals are well matched. Importantly the DIA isolation window schema and the MS2 level acquisition greatly reduced the noise background from a full mass-range MS1 level acquisition. These factors together ensure that the pSILAC-DIA data has a significantly better quantitative accuracy and reproducibility than the direct MS1-based quantification (even at 120k) (as compellingly shown in Appendix **Figures S3** and **S4**), especially for early pulse-chase time points which are known to heavily impact the protein turnover calculation (Claydon & Beynon, 2012).

In summary, the pSILAC-DIA quantification successfully improved the quantitative accuracy and reduced the MS data procession difficulty in pSILAC experiment, as compared to pSILAC-MS1 approach.

pSILAC-DIA vs. pSILAC-MS1 and pSILAC-TMT

One interesting advantage of pSILAC-DIA over both pSILAC-MS1 and pSILAC-TMT is the availability of multiple quantitative data points. This is because, for a given peptide (an H/L pair), very limited quantitative features can be obtained for pSILAC-MS1 (that are based on the precursor pair) or pSILAC-TMT (that are based on the reporter ion ratio from those identified MS2 or SPS-MS3 scan event). However, in pSILAC-DIA, all the high-resolution fragment ions of the peptide can be used for quantifying H/L ratios. Although normally the filtered top 3-6 fragment ions are already enough to determine the ratio, potentially almost all fragment ions can be used as replicated information, yielding a more robust estimation of H/L ratio. Please see **Appendix Figures S5** and **S6** for the further illustration. Although not shown in the current study, more ions could also potentially facilitate the turnover calculation for different peptidofoms (i.e., the same peptide sequences with different post-translational modifications and modification combinations).

pSILAC-DIA vs pSILAC-TMT

The hyper-multiplexing approach combining “pSILAC” and “MS2-tag based quantification” is not actually new, and has been already proposed in 2010 in which

pSILAC labeling was combined with iTRAQ quantification (i.e., pSILAC-iTRAQ) (Jayapal et al, 2010; Hinkson & Elias, 2011). Recently, pSILAC-TMT has emerged as a powerful MS based tool to measure protein turnover (Welle et al, 2016; Savitski et al, 2018; Zecha et al, 2018), especially because the SPS-MS3 quantification was shown to effectively reduce co-isolation issue in TMT measurements (McAlister et al, 2014; Zecha et al, 2018). Furthermore, there are more quantitative channels from TMT tags and better MS instruments available supporting high-resolution measurements needed for TMT quantification. Indeed, the multiplexity of pSILAC-TMT (Welle et al, 2016; Zecha et al, 2018) impairs the missing value problem and decreases the number of sample injections required for a pSILAC experiment.

However, in our hands pSILAC-DIA achieved comparable proteome coverage to pSILAC-TMT with a similar total MS measurement time. This could be due to the facts that e.g., there is no fractionation needed in pSILAC-DIA and that extra time for MS2 scans (that are not used for quantitation) was spent during SPS-MS3 TMT analysis. The comparable coverage between DIA and TMT approaches is consistent to a previous report comparing the performance of the two (Muntel et al, 2019). Besides the reduced ion complexity and more quantitative data points mentioned above (**Appendix Figures S2, S5, and S6**), in particular, we also summarize other advantages of pSILAC-DIA over pSILAC-TMT below. **Therefore, we suggest all these considerations together make pSILAC-DIA a competitive, if not a better, method than pSILAC-TMT.**

- 1) Even the current most optimized version of SPS-MS3 based pSILAC-TMT seems to (only) provide a similar quantitation performance of turnover rates to pSILAC-MS1, but not better. This may be not surprising as SPS-MS3 only largely reduces the co-isolation problem but does not totally eliminate it (Muntel et al, 2019), which stems from the TMT design. Thus, in the previous publications, the pSILAC-TMT (MS2) generated less-precise turnover rates than pSILAC-MS1 (Welle et al, 2016), whereas the pSILAC-TMT (SPS-MS3) with a further bioinformatic correction dealing with the normalization bias (Zecha et al, 2018), makes the approach become close or comparable to pSILAC-MS1 results in precisely determining the H/L ratios and thus, k_{loss} .

In contrast, again, with **Appendix Figures S3 and S4**, we show compelling results that pSILAC-DIA can generate significantly better quantification accuracy and precision than pSILAC-MS1 (see above).

- 2) The problem with large-scale, multi-batch TMT data was recently reported by experienced labs (Brenes et al, 2019), suggesting the pSILAC-TMT approach might not be ready to be used by every lab, and importantly, to be used to compare protein turnover across multiple cell lines (like our application shown in this study) and conditions. Indeed, false positives, batch effects, missing values, and other issues were reported in a multi-batch TMT experiment despite the use of SPS-MS3 (Brenes et al, 2019).
- 3) Experimental design flexibility. In a pSILAC experiment, a 10-time point sampling during pulse-chase process does not have to be always necessary (10 TMT channels were used in (Zecha et al, 2018)). More importantly, the pSILAC-DIA offers a greater flexibility in trouble-shooting a proteomic experiment (e.g., if one of the samples has some technical or experimental problem, in pSILAC-DIA workflow

the problematic sample can be easily re-processed and measured with new data combined, whereas in pSILAC-TMT the whole sample set has to be re-measured).

- 4) **Machine flexibility.** Currently, the SPS-MS3 seems to be essential for pSILAC-TMT to ensure the quality of turnover estimation to be comparable to pSILAC-MS1 (Zecha et al, 2018). This function, however, is only provided in Fusion-type Orbitrap analyzers from Thermo Scientific. pSILAC-DIA has a much broader application potential in this regard.
- 5) pSILAC-DIA might request **lower cost** for proteomics labs, considering the cost of TMT reagents for lots of samples.
- 6) Potential of pSILAC-DIA in **analyzing peptide PTM** isoforms (peptidoforms). DIA-MS includes extra information of the perfect co-elution behavior between all fragment ions of the same peptide isoform along the liquid chromatography. This feature will offer much higher ability in discriminating SILAC H/L ratios for discriminating PTM isoforms (Rosenberger et al, 2017). Instead, in pSILAC-TMT, those different PTM isoforms of the same peptide backbone eluting together (or, arising from other pSILAC channels of a different time point) can be easily co-isolated for MS2 and SPS-MS3 analysis, and thus, interfere with each other. We want to communicate with this reviewer that relevant phosphoproteomic datasets have been generated in the Liu lab that demonstrate this point.
- 7) Potential of analyzing heavy and light signals separately for **non-steady state measurement**. Although for the protein turnover measurement a general assumption is applied that cells are growing in the **steady state** (i.e., the sum of heavy and light signals is stable during labeling), DIA data provides the possibility to directly and robustly analyze heavy and light version of proteins separately. This ability will be useful for studying dynamic processes e.g., systems under a perturbation. To achieve such a task, pSILAC-TMT analysis would have to rely on a complex statistical approach (Savitski et al, 2018).

We now include the above discussions as a **Supplementary Note 3** in the **Appendix**.

Figure R1

Figure R1: Moderate correlation between protein turnover time determined by pSILAC-DIA and pSILAC-TMT.

is 0.51 between two datasets (**Figure R1-A**), whereas within our pSILAC-DIA dataset, the

As a final additional point, we compared the $T_{1/2}$ time ($T_{1/2} = \ln(2)/k_{\text{loss}}$) between our data measured by pSILAC-DIA and by pSILAC-TMT in Zecha et al. We selected this dataset because a similar proteome coverage was achieved in the same cell line. We found the correlation

correlation is much higher between HeLa CCL2 and HeLa Kyoto cells (**Figure R1-B**, Spearman's $\rho=0.85$). Interestingly, in Zecha et al, the authors also reported the correlation of protein half-lives derived from their pSILAC-TMT approach to three previous datasets generated using pSILAC-MS1 and found the correlation was 0.51-0.53. We therefore conclude that the protein turnover determined by either of both approaches also correlated as well with already published protein rates by pSILAC-MS1 or SILAC-TMT as these literature data correlated among each other.

In addition, I have a few minor points, as follows:

- Please explain the use of the term "loss rate" rather than "degradation rate", and please document the pulse-labelling better. I understand the samples had been described before, but since it is a paper about loss / turnover rates it would be good to know what the labelling time points were etc...

> The term "loss rate" (i.e., k_{loss}) is used to address the rate of loss of the unlabeled fraction of a peptide/protein in a dynamic pSILAC experiment. First, the relative isotope abundances (RIA) of the light peptide across several time points is calculated, and then an exponential curve fitting is performed to estimate the k_{loss} value. We now clarify these in **Methods (Line 23, Page 23, Line 1, Page 24)**.

Following this request, we now significantly extended the description of the method and documented experimental design of the pSILAC experiments in the **Methods** section and in a new figure (**Appendix Figure S1**). Furthermore, in the revised paper, we decided to consistently use either **protein degradation** (when we describe biological processes) and k_{loss} (when we refer to the parameter, as a proxy to *protein degradation rate*)

- Differential turnover of protein isoforms: How can you be sure to have different turnover rates for different protein isoforms? Presumably most of these measurements compare single peptides? If so, how do you know that's not just a difference between the peptides of the same protein? In general in these assays there appears to be quite a range of turnover rates identified by different peptides for the same protein (e.g. see Welle et al)

> Thanks for this question. We agree that addressing the degradation of individual splicing isoforms is a difficult task. Therefore, we referred to the statistical approach and statistics, in which we first selected only those protein isoforms with at least two unique peptide sequences with a valid k_{loss} value. Then, we performed a statistical test ($p < 0.05$; t-test or ANOVA followed by pairwise comparison by Tukey honest significant differences test), and additional fold-change cutoff was performed requiring \log_2 fold change of at least 0.32 (for at least one pairwise comparison in the case of the ANOVA results).

Following the comment, we now also performed more variability analyses on peptide entries (see **Appendix Supplementary Methods** and **Appendix Figure S7**). Provided the fact there might be a large variability of degradation rates for different peptides of the same protein, we further supported our observations by addressing the variability of peptides within genes, within protein AS isoform groups and between protein AS groups of the same gene.

- 1) Using the peptide level kloss data, we estimated a standard deviation of all peptide log₂ kloss values assigned to each gene (n = 2,390, n_{peptides} > 1) and to each protein AS group (n = 3,848, n_{peptides} > 1). We further visualized distributions of the standard deviations for genes and protein AS groups, and using Wilcoxon test, we tested whether the observed difference was statistically significant. This analysis was performed separately for the HeLa 1 log₂ kloss value, for HeLa Kyoto average log₂ kloss value, and for the HeLa CCL2 average log₂ kloss value.
- 2) Similar to a gene/protein-specific correlation between different layers, we calculated a correlation coefficient (Spearman's rho) between peptide log₂ kloss profiles across all HeLa cell lines. For every protein AS group (with at least two peptides quantified, n = 3,848), the correlations for all binary peptide comparisons were calculated (i.e., within- protein AS groups). For every gene (with two to four protein AS groups quantified with at least two peptides, n = 1,477), the correlations were calculated for all binary peptide comparison between peptides corresponding to different AS isoforms of the same gene (i.e., between- protein AS groups). We then visualized the distribution of all within- and between- protein AS groups correlations, and estimated statistical significance of the difference using Wilcoxon test. The same analysis was performed for a subset of 30 genes and the corresponding 60 protein AS groups for which we reported differential degradation between HeLa CCL2 and Kyoto ($p < 0.05$, only genes with 2 protein AS groups were used for this analysis).

In summary, these plots thus support the notion that the peptides come from different isoforms, which cannot be explained alternatively as they are only from different parts of the same protein. The analyses are now summarized in the **Appendix Supplementary Methods** and **Appendix Figure S7**.

- The title gives the impression that both mRNA and protein turnover were measured

> We now adjust the title to “**Isoform-resolved correlation analysis between mRNA abundance regulation and protein-level degradation**”.

We hope this removes the impression that mRNA degradation is measured.

- Page 4, line 9: I think you mean "sequence" database not "proteomic" database?

> We agree and changed it to “protein sequence” database.

- 6,552 proteins using a single-shot analysis seems amazing. For how many proteins do you actually have kloss rates and in how many samples? How many missing values are there? Essentially, please document the dataset a little better

> We now include a supplementary table for all k_{loss} results across all samples (**Table EV1**). We further summarized our results and the overlap between peptides and protein AS

groups that were quantified in the label-free DIA measurements and for which we successfully estimated the k_{loss} values in all samples and provide these data as below.

	peptide expression				peptide k_{loss}			
	peptides		protein AS groups	genes	peptides		protein AS groups	genes
full profiles, unique genes	51,291	1 pept	11,988	5,434	24,275	1 pept	6,994	3,518
		2 pept	7,388	4,179		2 pept	3,848	2,390

Figure R2

Figure R2: The number of peptides, protein AS groups, and genes quantified in all HeLa cell lines.

We also add a note in Results e.g., “This translates to 6,994 protein AS groups (3,518 gene symbols collapsed) quantified with a degradation rate in every of the 12 HeLa samples” (**Line 17-18, Page 7**).

- Fig. 2D-G: No axis labels

> We now add the labels to the figure (now **Figure 3D-G**).

- The discussion is overstating the claims of novelty of the study quite considerably, e.g. the first (page 17, line 11) and second (line 13) points are not specific for this paper.

> We have rewritten lots of parts of the Discussion (e.g. **Para 1, Page 16**). For example, we put more focus on the pSILAC-DIA methodology. In particular for this comment, we have removed the first and second points in the discussion. This also makes the discussion more concise.

Reviewer #2:

Salovska et al. addressed the question of how protein levels, mRNA levels and protein turnover relate to each other. A single gene can give rise to many transcript isoforms, which then are translated into different proteoforms, very often with unique functions within the cell. In most previous integrated proteomics studies the regulation of gene expression of different proteoforms was not directly assessed. Salovska et al. generate an impressive data set that indeed allows the study of the interplay between mRNA levels, protein levels and protein with a special focus of their analysis to a quite large panel to alternative spliced isoforms, which provides significant novelty. Moreover, the peptide centric analysis by DIA of pulsed SILAC (pSILAC) data is a very nice new way to analyze pSILAC data and could be extremely useful for the community. They applied their novel approach to gain new insight about gene expression across about a dozen different strains of HeLa cell lines and reported a variety of different regulatory patterns between cells and for different gene groups. All in all I think the authors generated a very impressive and intriguing data set that definitely would be of great use for the gene expression community and could potentially also serve as a blueprint for studying different layers of gene expression. However, I think before acceptance of the manuscript can be considered several issues absolutely need to be addressed.

> We thank for the positive comments.

The main issues are the following:

1. It is hard to understand if the authors mean protein degradation, protein loss or protein turnover. Although the first two terms the same, the third is not really the same as this is affected by protein production and protein loss, but this is actually what I think they are measuring and should be using. However, the authors constantly push the term Kloss, which would indicate that they only look at protein degradation and therefore should be only dependent on the light signal. Yet, if one reads through materials and methods it seems that the authors calculate Kloss as $L/(H+L)$, but that is not really the loss of protein, but the percentage of L of the total protein and depends on H and therefore protein production. Again true protein degradation should only look at the L signal over time (and use the H+L signal just for normalization for same sample input). The way the authors calculate Kloss profoundly affects the interpretation of the results. For example the loss of

L (protein degradation) could be the identical between the different cell types, but the protein production (there for the H signal) could be different between the cell types (e.g. mRNA is upregulated and more protein is produced), but the way the authors calculate protein degradation and just by H being different it would look like a higher Kloss for the authors, although it is just higher turnover not higher loss. It should be noted that in such a case as described there should also be a higher amount in total protein. So in reality that would not be buffering but an amplification or neither or as for example if mRNA is up I would expect H to go up also, without L changing. So that would just be all driven by mRNA change, but in their formula it would look like that Kloss decreases to buffer, which it actually does not. So especially conclusion if something is amplified or buffered have to be taken with caution with the way the authors calculate Kloss.

It might easily be that I misunderstand something and I am missing something obvious, but if I got it correctly how the authors calculated Kloss then at the present state of my understanding I do not agree with many of their conclusions. This could be easily addressed by reanalysis of the data (e.g. calculating the real protein loss).

> We regret the misleading usage of *protein turnover* and *protein degradation*. We believe a clear reasoning of pSILAC based protein turnover measurement is helpful eliminate this confusion.

Firstly, in the experiments using pSILAC based turnover determination, we are working with an important assumption, i.e., the cells are respectively maintained in a **steady state** (i.e., without any perturbation), so that for a given protein with known concentration, the degraded and synthesized protein copies are balanced. Under this assumption (which is widely used in previous pSILAC based experiments, (Claydon & Beynon, 2012)), the respective determination of the protein-specific k_{loss} within each HeLa cell line directly estimates protein turnover behavior in that cell line. We now provide a supplementary figure as the first **Appendix Figure (Figure S1)** to display the basic concepts and this assumption before the whole analysis being presented in the paper.

Secondly, this reviewer is correct that when two cell line (or cell types) are compared, we only compared change of “degradation rate” but not the “turnover rate”. To clarify, a relevant concept is protein synthesis rate. The simplest model of protein turnover assumes that synthesis is a zero-order process and degradation is a first-order process. This means, the rate of synthesis ($K_{synthesis}$) has the units of molecules, or copies per cell, whereas degradation, being fractional removal from the pool, has the dimensions of time⁻¹ (i.e., $K_{degradation}$). For a protein with the copies per cell (i.e. [P] as the number of protein molecules),

$$dP/dt = K_{synthesis} - [P] \times K_{degradation}$$

Because we assume the protein pool size is constant under steady state, i.e., $dP/dt = 0$, $K_{synthesis}$ is determined only by protein amount [P] and degradation rate. “Turnover rate” certainly also involves $K_{synthesis}$. However, for previous turnover studies, the major object is to determine the first-order rate constant for degradation of each protein of interest, i.e., the so-called “turnover rate” (Claydon & Beynon, 2012). Thus the concepts of protein turnover

and protein degradation, as two terms, were often used together (mixed) in previous studies.

Thirdly, following, we agree that when multiple cell lines are compared, one should stick to the term of protein degradation (i.e., [P] can be different between cell lines). Therefore, in the revised paper, **we are now keeping the consistent usage of protein degradation** (when we describe biological processes) and k_{loss} (when we refer to the parameter, as a proxy to $K_{\text{degradation}}$, See below) throughout the manuscript, for relative analysis between cells.

Last but not the least, as a biological point, we are studying the correlation between mRNA regulation and protein degradation regulation. We have move the previous **Figure EV4A** to a main Figure (**new Figure 5D**) to illustrate that the protein variability indeed reduced with better mRNA- k_{loss} , demonstrating mRNA- k_{loss} is indeed a good measure for buffering attempts mediated by protein turnover. This is consistent to previous reports suggesting that the adaption of translation rates does not explain buffering of proteins (Albert et al, 2014; Bader et al, 2015). We now modified the introduction and result text to make this clearer (e.g., **Line 30, Page 2; Line 9-16, Page 11**).

Other actions: Furthermore, we add the below text describing the detailed process of k_{loss} calculation and assumptions into the **Methods** to improve the readability (**Line 23 Page 22; Line 1 Page 23**). Please see text pasted below.

At each time point, the amount of heavy (H) and light (L) precursor was extracted and used to calculate the relative isotopic abundance RIA_t .

$$RIA_t = \frac{L}{L + H}$$

This is analogous to Pratt et al (Pratt et al, 2002). The value of $RIA(t)$ is time dependent, as unlabeled proteins are replaced with heavy-labeled proteins during the course of the experiment. This is due to dilution of the cells as well as intracellular protein turnover, where the rate of loss can be modeled as an exponential decay process.

$$RIA_t = RIA_0 \cdot e^{\{-k_{\text{loss}} \cdot t\}}$$

where RIA_0 denotes the initial isotopic ratio and k_{loss} the rate of (hourly) loss of unlabeled protein. We assumed $RIA_0 = 1$, as no heavy isotope was present at $t = 0$, thus the value of RIA_t will decay exponentially from 1 to 0 after infinite time and used nonlinear least-squares estimation to perform the fit. As discussed in Pratt et al (Pratt et al, 2002), these assumptions may reduce measurement error especially at the beginning of the experiment, where isotopic ratios are less accurate owing to the low absolute number of heavy precursor ions where our new pSILAC-DIA strategy is helpful. The k_{loss} for each protein isoform groups was computed as the median of all peptide-level rates. We excluded proteins quantified in a single time point only and increasing isotope ratio over time.

As previously documented (Liu et al, 2019), to avoid the possible calculation issues due to light amino acid recycling (Boisvert et al, 2012) and inaccurate cell doubling time measurement in different HeLa cells, we use $\log_2 k_{\text{loss}}$ as a proxy estimate to protein degradation rate (Liu et al, 2019).

2. Data presented in Figure 4 (and also partly Figure 2): all the correlation - what has been correlated with what for each gene. For example in 4A in the mRNA to Kloss plot - did you compare the mRNA intensity value to the Kloss value for each gene and this for each cell line separately and these 12 data points gave the spearman rho for each gene? This was not completely clear to me - please clarify? This is also true for Figure 2 - not really 100% clear on what all the correlations are based on - e.g. Figure 2H to 2J. It is essential to understand how that was calculated exactly. I might have missed it, but did also not find that information in materials and methods.

> We now clarify that this is across- cell, gene-specific correlation between mRNA and Kloss. This means that, for every gene, each cell line will have a mRNA abundance value and a Kloss value, which forms a “data point” in the correlation analysis (**Figure R3**). Therefore, for each gene, there will be 12 data points from which we can calculate a correlation coefficient – and this correlation coefficient is calculated for each gene. This correlation is exactly what this reviewer interprets (**Figure R3**).

Figure R3

We now clarified this with the new descriptions in the **Methods section (Para 1, Page 25)** and also in the **figure legends** separately for **Figure 4** (now **Figure 5**) and **Figure 2H to 2J** (now **Figure 3H to 3J**).

Figure R3: Gene/protein specific correlation between multiple HeLa cell line strains.

More minor issues:

In general the manuscript is not an easy read and sometimes I was not sure what the authors mean. In particular:

- Page 4, Line 15 to 17: that sentence seems to have important information but the way it is presented it is hard to get. Maybe elaborate in one or two sentences more.

> We tried to improve the readability with the new manuscript. Particularly here, we rewrite here.

“We have developed an approach differentiating the major and unique AS isoforms for each AS group. Based on the abundance of the major transcript AS isoform, we confirmed that protein abundance is generally correlated to transcript AS levels in a spliceosome-disrupted system.” (**Page 3, Line 14-18**).

- Page 6, third paragraph, Lines 15 to 27: based on the description I am not 100% sure what you did. Did you actually spike in peptides or just use the light signal already present from the pSILAC?

> We used the light signal in the pSILAC-DIA data. The ISW is just the name of the data analysis workflow introduced for a particular SRM experiment (e.g., in heavy labeled cells where the light spike-in peptides can be used to save peptide synthesis cost). Herein we borrowed this concept. We have now improved the description here to clarify (**Line 11-24, Page 5**).

“To compensate for the limitations, we herein adopted the peptide identification strategy used in the inverted spike-in workflow (ISW) (Reiter et al, 2011) for analyzing pSILAC-DIA data (**Fig 1B**). ISW was initially introduced in a particular type of SRM experiments where the “light” synthetic peptides were spiked as standards. This means, in ISW the peptide detection scoring process is only based on the q-values of “light” precursors. Thus, in pSILAC-DIA dataset ISW will maximize reliable detection of newly synthesized “heavy” protein in the early time points during labeling, when the “heavy” signals are much lower than those “light” ones of pre-existing protein copies. Indeed, ISW increases the number of heavy-to-light ratios by 30.6% and a 14.7% in the dataset with 1-hour and 4-hour pSILAC labeling (see **Appendix**) (Reiter et al, 2011). The ISW is now available in a new version of Spectronaut software (Bruderer et al, 2015) by which both “heavy” and “light” MS assays can be reversibly generated based on either DIA or DDA datasets or both. Furthermore, the heavy-versus-light elution was assembled before DIA identification, so that no post feature alignment (Rost et al, 2016) is required (see **Appendix** for a step-by-step protocol for pSILAC data analysis and related data assessment). ”.

- Page 13, Line 28 and 29: "less stabilized by turnover" - not sure what that means - degraded faster? Turned over faster?

> Thanks. We now changed the sentence to “whereas the transcriptional changes of 19S proteins are not further regulated by protein degradation...”. (**Line 5, Page 13**)

- Page 16, Line 15 and 16: The author write rightly that protein turnover is "denotes the balance between protein synthesis and degradation for a final product.", but are constantly talking about k_{loss} . What is it now?

> We deleted this sentence and replace it with (**Line 12, Page 15**) “In a steady state within a cell system, the stable protein concentration is the result of the balance between protein synthesis and degradation.” in the revision. Please also refer to major point 1 from this Reviewer. To keep consistent, we are using protein turnover and k_{loss} throughout the manuscript. Again, under the assumption of steady-state, synthesis and degradation are balanced.

- Page 18, Line 26 till 28: I do not really fully understand that sentence - maybe rephrase?

> We apologize for the complexity of this sentence. We now rewrite this sentence.

“Importantly, we expect that many genes involved in these processes can serve as a list to predict the “preferably regulated” turnover events, especially when the up- or down-regulations of mRNA are determined between different steady states.” (**Line 11, Page 17**).

Other minor issues:

- Page 9, Line 9 to 11: Maybe I am missing something obvious, but I would actually suggest that the transcriptional induction leads to faster protein turnover not a surprise as more protein is produced upon induction and therefore also turnover increased.

> We agree. Our writing was indeed misleading here. The point here is about absolute and relative mRNA- k_{loss} correlation provide different biological insight. Our data suggests that the relative transcriptional changes are mainly subjected to post-translational buffering by relative protein degradation rate (buffering means the proteome variation is reduced from the mRNA level). With our relative analysis, we were able to study this at the proteome / genome level.

Following this suggestion, we have significantly rewritten the paragraph here (**Line 24, Page 8**).

“The absolute mRNA- k_{loss} correlation was determined to be $\rho = -0.14$ and -0.17 (**Fig 3D&E**; $n = 885$ and $2,895$, $p < 0.001$) in UQ and SM proteins, and the absolute protein- k_{loss}

correlation is even more negative ($\rho = -0.31$ for UQ and -0.29 for SM). Thus, protein degradation seems to be slower for higher gene expression.”

- Page 9, Line 17 to 20: These correlations are extremely weak and I think that these kind of conclusions like that protein turnover buffers mRNA variation are extremely strong for such weak correlations, even if significant (which is not surprising with the number of data points).

- Page 10, Line 4 to 6: Again the values are very small and it is hard to believe that this is really the case at such an effect size.

> Indeed, these are weak, but significant correlations. Exactly because of this, we dissect these correlations further in the previous **Figure 4** (now **Figure 5**). Importantly, the relative fold-change correlation is also back up by the across-cell, gene specific correlation histogram (**Figure 5A**). We also reinforced the argument with **Figure 2H-J** (now **Figure 3H-J**). This weak correlation is consistent to previous studies including ours (despite at a smaller scale). We now cite these papers (**Para.1, Page 15, Line 16, Page 17**).

Herein we further provide **Figure R4**, where we plot the mRNA- k_{loss} correlation (again, gene-specific, across-cell line correlation) distributed to protein complex participation. This kind plot was already used in our previous papers, where proteins that are known to be the subunits of a stable protein complex (e.g., according to CORUM annotation) are marked as “Complex_IN”, whereas other proteins without a protein complex annotation are marked as “Complex-OUT”. Because protein complex stoichiometry is a general mechanism that is well known for protein buffering, the mRNA- k_{loss} is indeed a good measure and a good study object based on our analysis of thousands of pair across multiple cell lines. The better mRNA- k_{loss} correlations result in more constraints at the protein level and thus worse mRNA-protein correlations (**Figure R4**). Our proteome coverage and analysis scale are already state-of-the-art. This also significantly reinforced previous reports suggesting that the adaption of translation rates does not explain buffering of proteins (Albert et al, 2014; Bader et al, 2015).

We now emphasized the fact that these numbers are “slight but significant” (**Line 3, Page 9**). Moreover, by no means we want to claim that all mRNA changes are buffered. We also improved the discussion of biological significance (**Para. 2 Page 17**).

Figure R4

Figure R4: Gene/protein specific correlation resolved to protein complex subunits and other proteins.

Page 12, Line 13 and 14: This is simply not true as far as I understand they are looking at turnover and this influenced by production and degradation and therefore higher production at the same rate of degradation can also lead to faster turnover and it is not necessary degradation. They should put this then at least in relation to total protein (as this should be higher in such an example).

> This is related to major point 1-2 above. However, we agree that we should use more accurate terms, especially we are studying multiple cells which is different to the previous pSILAC studies.

The sentence now reads like “This skewed distribution again indicates mRNA regulation attempts are more often prone to be buffered rather than to be amplified by protein degradation.” (Line 29-31, Page 11).

- Page 13, Line 10 to 12: how does the total protein level look for those categories. Also went up - if so no buffering necessarily, but potentially even amplification.

> This is an important point. We now clarify and stress that the “buffering” of protein level actually is independent of final protein abundance fold-change. For example, if one protein should have been upregulated by 10 times due to transcriptional regulation – with “buffering” from mRNA variation it is upregulated by only 4 times. This is a both significant buffering and also a significant, “final” upregulation. Therefore, we suggest that the buffering can be inferred with the CV of protein levels between cells to compare different proteins (the new **Figure 5D**), and more importantly the mRNA regulation and protein degradation correlation in whole **Figure 5** – which indicates the buffering result

and attempt through protein degradation control. We tried to improve the clarification of these concepts throughout the paper in this revision.

Again, I want to stress that this is actually a very nice data set and I do absolutely think publication of it in MSB would be of great interest to the proteomics and gene expression community, but at the current state certain things need to be addressed first. Having said that I am positive that the authors can address most of my concerns.

> We hope our new analysis, clarification, and the significantly rewritten manuscript now are acceptable to this reviewer.

Reference

Albert FW, Muzzey D, Weissman JS, Kruglyak L (2014) Genetic influences on translation in yeast. *PLoS Genet* **10**: e1004692

Bader DM, Wilkening S, Lin G, Tekkedil MM, Dietrich K, Steinmetz LM, Gagneur J (2015) Negative feedback buffers effects of regulatory variants. *Molecular systems biology* **11**: 785

Boisvert FM, Ahmad Y, Gierlinski M, Charriere F, Lamont D, Scott M, Barton G, Lamond AI (2012) A quantitative spatial proteomics analysis of proteome turnover in human cells. *Molecular & cellular proteomics : MCP* **11**: M111 011429

Brenes A, Hukelmann J, Bensaddek D, Lamond AI (2019) Multibatch TMT Reveals False Positives, Batch Effects and Missing Values. *Molecular & cellular proteomics : MCP* **18**: 1967-1980

Bruderer R, Bernhardt OM, Gandhi T, Miladinovic SM, Cheng LY, Messner S, Ehrenberger T, Zanotelli V, Butscheid Y, Escher C, Vitek O, Rinner O, Reiter L (2015) Extending the limits of quantitative proteome profiling with data-independent acquisition and application to acetaminophen-treated three-dimensional liver microtissues. *Mol Cell Proteomics* **14**: 1400-1410

Claydon AJ, Beynon R (2012) Proteome dynamics: revisiting turnover with a global perspective. *Molecular & cellular proteomics : MCP* **11**: 1551-1565

Forshed J, Johansson HJ, Pernemalm M, Branca RM, Sandberg A, Lehtio J (2011) Enhanced information output from shotgun proteomics data by protein quantification and peptide quality control (PQPQ). *Molecular & cellular proteomics : MCP* **10**: M111 010264

Franken H, Mathieson T, Childs D, Sweetman GM, Werner T, Togel I, Doce C, Gade S, Bantscheff M, Drewes G, Reinhard FB, Huber W, Savitski MM (2015) Thermal

proteome profiling for unbiased identification of direct and indirect drug targets using multiplexed quantitative mass spectrometry. *Nature protocols* **10**: 1567-1593

Hinkson IV, Elias JE (2011) The dynamic state of protein turnover: It's about time. *Trends Cell Biol* **21**: 293-303

Jayapal KP, Sui S, Philp RJ, Kok YJ, Yap MG, Griffin TJ, Hu WS (2010) Multitagging proteomic strategy to estimate protein turnover rates in dynamic systems. *Journal of proteome research* **9**: 2087-2097

Liu Y, Borel C, Li L, Muller T, Williams EG, Germain PL, Buljan M, Sajic T, Boersema PJ, Shao W, Faini M, Testa G, Beyer A, Antonarakis SE, Aebersold R (2017) Systematic proteome and proteostasis profiling in human Trisomy 21 fibroblast cells. *Nature communications* **8**: 1212

Liu Y, Mi Y, Mueller T, Kreibich S, Williams EG, Van Drogen A, Borel C, Frank M, Germain PL, Bludau I, Mehnert M, Seifert M, Emmenlauer M, Sorg I, Bezrukov F, Bena FS, Zhou H, Dehio C, Testa G, Saez-Rodriguez J et al (2019) Multi-omic measurements of heterogeneity in HeLa cells across laboratories. *Nature biotechnology* **37**: 314-322

Mathieson T, Franken H, Kosinski J, Kurzawa N, Zinn N, Sweetman G, Poeckel D, Ratnu VS, Schramm M, Becher I, Steidel M, Noh KM, Bergamini G, Beck M, Bantscheff M, Savitski MM (2018) Systematic analysis of protein turnover in primary cells. *Nature communications* **9**: 689

McAlister GC, Nusinow DP, Jedrychowski MP, Wuhr M, Huttlin EL, Erickson BK, Rad R, Haas W, Gygi SP (2014) MultiNotch MS3 enables accurate, sensitive, and multiplexed detection of differential expression across cancer cell line proteomes. *Analytical chemistry* **86**: 7150-7158

Muntel J, Kirkpatrick J, Bruderer R, Huang T, Vitek O, Ori A, Reiter L (2019) Comparison of Protein Quantification in a Complex Background by DIA and TMT Workflows with Fixed Instrument Time. *Journal of proteome research* **18**: 1340-1351

Pratt JM, Petty J, Riba-Garcia I, Robertson DH, Gaskell SJ, Oliver SG, Beynon RJ (2002) Dynamics of protein turnover, a missing dimension in proteomics. *Molecular & cellular proteomics : MCP* **1**: 579-591

Reiter L, Rinner O, Picotti P, Huttenhain R, Beck M, Brusniak MY, Hengartner MO, Aebersold R (2011) mProphet: automated data processing and statistical validation for large-scale SRM experiments. *Nat Methods* **8**: 430-435

Rosenberger G, Liu Y, Rost HL, Ludwig C, Buil A, Bensimon A, Soste M, Spector TD, Dermitzakis ET, Collins BC, Malmstrom L, Aebersold R (2017) Inference and

quantification of peptidofoms in large sample cohorts by SWATH-MS. *Nature biotechnology* **35**: 781-788

Rost HL, Liu Y, D'Agostino G, Zanella M, Navarro P, Rosenberger G, Collins BC, Gillet L, Testa G, Malmstrom L, Aebersold R (2016) TRIC: an automated alignment strategy for reproducible protein quantification in targeted proteomics. *Nat Methods* **13**: 777-783

Savitski MM, Zinn N, Faelth-Savitski M, Poeckel D, Gade S, Becher I, Muelbaier M, Wagner AJ, Stroemer K, Werner T, Melchert S, Petretich M, Rutkowska A, Vappiani J, Franken H, Steidel M, Sweetman GM, Gilan O, Lam EYN, Dawson MA et al (2018) Multiplexed Proteome Dynamics Profiling Reveals Mechanisms Controlling Protein Homeostasis. *Cell* **173**: 260-274 e225

Welle KA, Zhang T, Hryhorenko JR, Shen S, Qu J, Ghaemmaghami S (2016) Time-resolved Analysis of Proteome Dynamics by Tandem Mass Tags and Stable Isotope Labeling in Cell Culture (TMT-SILAC) Hyperplexing. *Molecular & cellular proteomics : MCP* **15**: 3551-3563

Zecha J, Meng C, Zolg DP, Samaras P, Wilhelm M, Kuster B (2018) Peptide Level Turnover Measurements Enable the Study of Proteoform Dynamics. *Molecular & cellular proteomics : MCP* **17**: 974-992

2nd Editorial Decision

30th January 2020

Thank you again for sending us your revised study. We have now heard back from the two referees who were asked to evaluate your study. As you will see below, the reviewers acknowledge that the study has significantly improved as a result of the performed revisions. They raise however a few remaining concerns, most of which can be addressed by text modifications, which we would ask you to address in a minor revision.

REFEREE REPORTS

Reviewer #1:

The authors have added several new analyses and figures, and as a result the manuscript has been significantly improved. For example, the new supplementary note 3 comparing the new pSILAC-DIA method to existing methods is very helpful. I also think the Discussion section reads better now. I think it would still be good to make the manuscript text a little bit clearer overall, for example this sentence in the abstract is hard to understand: "The dataset demonstrates that specific biological processes, cellular organelles, subunits of organelles, and individual protein isoforms of same genes could have distinctive degradation rate and the corresponding buffering or concerting protein turnover control across cancer cell lines." Also, the term subunits is typically used for protein complexes, whereas I think this sentence refers to spatial compartments within organelles (e.g. mitochondrial matrix vs inner membrane).

Reviewer #2:

I want to thank the authors that they took my concerns seriously and to provided such detailed answers. Moreover, I think the revised manuscript is much improved and easier to understand. The authors addressed most of my concerns sufficiently and in general I fully support acceptance.

However, it would be great if the authors could still address the points below (especially point 1):

1. I do still have a problem with the definition of Kloss. I think I do understand all the assumptions the authors make (protein levels are at steady state and therefore production equals protein loss), but still think that the way Kloss is calculated it does not really present protein degradation as the authors claim. It might be an issue of semantics only and could still easily be my misunderstanding, but here is my main issue:

I still believe that defining Kloss as fraction of total protein is not an accurate proxy for protein degradation, but related to turnover as both labels H and L play a role. This is especially critical if we look at relative comparisons. As stated in the first round of reviews - my belief and definition of protein degradation is that this only depends on the loss of the L signal over time. In that sense it will be in relation to the L signal at time point 0h (L_{xh} / L_{0h}). I know the authors have a different definition, based also on literature, which is $L/(L+H)$. The term as the authors use it, is in my opinion directly related to protein turnover as it incorporates also the H signal. In the case of a relative comparison between the cell lines the H value and therefore the (H+L) can differ significantly if the protein is differently expressed, meaning that for example in the case when a protein is upregulated the denominator is always bigger and this would lead automatically to stronger Kloss if the numerator values actually stay the same between conditions.

Boisvert et al. (MCP, 2012) and Jovanovic et al. (Science, 2015) for example only use the loss of the "Medium Channel" to calculate protein degradation. The heavy channel signal (which is also the production channel in their case) is not used. Yes in both studies it is in reality the "M/L" ratio over time, but in this case L is really only a spike in channel that helps with the normalization.

Due to the good data quality provided by the authors here, no third channel should be necessary and they should be able to directly look at the loss of L relative to time point 0h to estimate the degradation rate, basically replicating what was done in Boisvert et al. (MCP, 2012) and Jovanovic et al. (Science, 2015).

Normally I would not make such a fuss about something that is admittedly more an issue of definition, but one of the main conclusion of the paper is that Kloss (which they say themselves resembles protein degradation) buffers mRNA increase. Although this conclusion might (and most likely is) right, I think that the way the authors come to it is misleading. I would like to see if it holds still true if the authors calculate protein degradation as in Boisvert et al. (MCP, 2012) and Jovanovic et al. (Science, 2015).

An additional test that should be included:

- The authors correctly claim that buffering means that for example a ten fold increase in mRNA could only lead to a four fold increase in total protein and this should rightfully be also called buffering, even so the expression level changes significantly between both conditions at mRNA and protein level.

If the authors conclusion is right that this buffering is really due to Kloss then I would like to see that proteins that show strong mRNA changes and seem buffered at the protein level indeed have a more significant kloss change than genes that have mRNA changes but the protein level changes correspond accordingly. The authors could for example bin genes based on the mRNA fold changes and then in each bin select a number of genes where the mRNA and protein changes correspond well and determine for these the Kloss to mRNA correlation and compare that to mRNA to Kloss correlation in the genes in the same bin where the mRNA changes seem buffered at the protein level.

- As a general note - any conclusions about absolute buffering have to be made cautiously as the absolute values of the relative changes matter and then the different dynamic range between RNA-seq and DIA pSILAC mass spectrometry could matter.

I have to say this is one of the cases where I wish I could talk to the authors directly to clarify. There might be a big misunderstanding on my side and I am sure that a few minutes of discussion would

clarify a lot.

A few very minor issues:

2. I don't understand the following conclusion on page 9 (last sentence is not clear):

"Correspondingly, the mRNA-protein correlation is high for differentially expressed proteins (median $\rho = 0.70$ and 0.66 , UQ and SM; Fig 3H), indicating protein level variability globally follow mRNA changes. Interestingly, only 10.96% (that is, 97 out of 885) and 12.61% (that is 365 out of 2,895) of UQ and SM proteins had a markedly regulated degradation rate (Fig 3J). To summarize, the protein degradation globally tunes protein levels but has significant buffering preferences. "

3. Explain the following conclusion on page 10. Is that because you see the general trend that high expressed genes have a lower kloss?

"However, the "complex I" proteins are much less abundant than "matrix" proteins. Thus, the dramatic degradation rate difference between the two may be largely ascribed to the absolute protein abundance difference (Fig EV3-B)."

4. Page 11 top - please show intensity distribution of protein measurements and RNA measurements for each quintal to make sure that the difference in CV is not due to different signal to noise ratios in each quintal.

"...the top 20% proteins (Q5) with highest mRNA-kloss correlation show lower coefficients of variation (CV) than the bottom 20% proteins (Q1), and both groups were significantly different from the rest of the data (Q2-Q4), confirming the modulation of protein levels by protein degradation-mediated buffering mechanisms (Kruskal-Wallis $p < 1e-10$; Fig 5D)."

5. Page 16: Small spelling error. I guess it should mean "and" and not "ad"

"Secondly, we used an mRNA abundance directed approach to identify and quantify signals to isoform groups."

6. Not sure what is meant by the second part of the sentence on page 16 (I guess the "whereas" confuses me a bit?)

"The transcript abundance-oriented detection of AS peptides enables the detection of exons only with appreciable read counts, whereas the transcript abundance-oriented quantification of AS enables a wider functional annotation. "

2nd Revision - authors' response

6th February 2020

Reviewer #1:

The authors have added several new analyses and figures, and as a result the manuscript has been significantly improved. For example, the new supplementary note 3 comparing the new pSILAC-DIA method to existing methods is very helpful. I also think the Discussion section reads better now. I think it would still be good to make the manuscript text a little bit clearer overall, for example this sentence in the abstract is hard to understand: "The dataset demonstrates that specific biological processes, cellular organelles, subunits of organelles, and individual protein isoforms of same genes could have distinctive degradation rate and the corresponding buffering or concerting protein turnover control across cancer cell lines." Also, the term subunits is typically used for protein complexes, whereas I think this sentence

refers to spatial compartments within organelles (e.g. mitochondrial matrix vs inner membrane).

> Author: Thanks. We have proof read the paper again to improve the text.

Also, for this particular sentence in the abstract, we now break it down and changed the wording of “subunits” (to “spatial compartments” as suggested). Because the two examples we focused on in the manuscript are in general regarded as “protein complexes”, we kept the usage of “subunits” in the main text where proteasome and mitoribosome are mentioned separately.

This sentence in the Abstract now reads like: **“The dataset revealed that specific biological processes, cellular organelles, spatial compartments of organelles, and individual protein isoforms of same genes could have distinctive degradation rate. The protein degradation diversity thus dissects the corresponding buffering or concerting protein turnover control across cancer cell lines.”**

Reviewer #2:

I want to thank the authors that they tool my concerns seriously and to provided such detailed answers. Moreover, I think the revised manuscript is much improved and easier to understand. The authors addressed most of my concerns sufficiently and in general I fully support acceptance. However, it would be great if the authors could still address the points below (especially point 1):

1. I do still have a problem with the definition of Kloss. I think I do understand all the assumptions the authors make (protein levels are at steady state and therefore production equals protein loss), but still think that the way Kloss is calculated it does not really present protein degradation as the authors claim. It might be an issue of semantics only and could still easily be my misunderstanding, but here is my main issue:

I still believe that defining Kloss as fraction of total protein is not an accurate proxy for protein degradation, but related to turnover as both labels H and L play a role. This is especially critical if we look at relative comparisons. As stated in the first round of reviews - my belief and definition of protein degradation is that this only depends on the loss of the L signal over time. In that sense it will be in relation to the L signal at time point 0h (L_{xh} / L_{0h}). I know the authors have a different definition, based also on literature, which is $L/(L+H)$. The term as the authors use it, is in my opinion directly related to protein turnover as it incorporates also the H signal. In the case of a relative comparison between the cell lines the H value and therefore the (H+L) can differ significantly if the protein is differently expressed, meaning that for example in the case when a protein is upregulated the denominator is always bigger and this would lead automatically to stronger Kloss if the numerator values actually stay the same between conditions.

Boisvert et al. (MCP, 2012) and Jovanovic et al. (Science, 2015) for example only use the loss of the "Medium Channel" to calculate protein degradation. The heavy channel signal (which is also the production channel in their case) is not used. Yes in both studies it is in reality the "M/L" ratio over time, but in this case L is really only a spike in channel that helps with the normalization.

Due to the good data quality provided by the authors here, no third channel should be necessary and they should be able to directly look at the loss of L relative to time point 0h to estimate the degradation rate, basically replicating what was done in Boisvert et al. (MCP, 2012) and Jovanovic et al. (Science, 2015).

Normally I would not make such a fuss about something that is admittedly more an issue of definition, but one of the main conclusion of the paper is that Kloss (which they say themselves resembles protein degradation) buffers mRNA increase. Although this conclusion might (and most likely is) right, I think that the way the authors come to it is misleading. I would like to see if it holds still true if the authors calculate protein degradation as in Boisvert et al. (Proteomics, 2012) and Jovanovic et al. (Science, 2015).

> Author: We believe this concern Reviewer 2 holds is indeed an issue of semantics and more "an issue of definition". In this revision, we now performed all the tests this reviewer asked (see **Supplementary Note 4** including four additional figures within the Note). However, before introducing the **Note**, we would like to simply state that, a) under the assumption of "steady state" (i.e., the sum of H+L remain constant over pSILAC labeling course), modeling L alone or modelling L/(L+H) barely have any practical difference (again, L+H is assumed to be constant), b) the RIA based k_{loss} measurement/determination is independent between cell lines. This means, the relative correlation analysis was performed "after" the respective k_{loss} calculation for each cell line. As the reviewer agrees, the RIA method was used in many previously published papers (Pratt et al, 2002; Doherty et al, 2009; Claydon & Beynon, 2012; Rost et al, 2016) analyzing steady state conditions (which did not use any medium signal - M). At the end of the day, we have to use either term "turnover rate" or "degradation rate" for our calculation. Please see more relevant discussions in our previous revision.

Nevertheless, we hope the new **Supplementary Note 4** (pasted below) expels this remaining concern and facilitates timely publication of our study. We also described the requested calculation in the **Methods** (the last paragraph, Page 21).

Supplementary Note 4: Comparison between the relative isotope abundance (RIA)-based and normalized light intensities (NLI)-based k_{loss} calculations

The RIA algorithm used in this study determines the protein turnover rate in the steady state considering both "light" (L) and "heavy" (H) intensities (see **Methods**). However, the high accuracy of our DIA-MS measurement actually enables measuring L and H peptide intensities separately in the pSILAC experiment. A simpler approach to determine a *de facto* protein degradation rate

would be to directly calculate the rate of loss from the L (*i.e.*, unlabeled peptide) intensities. Therefore, in this Supplementary Note, we performed such a calculation termed normalized light intensity (NLI)-based method (see **Methods**) and compared the results to results determined by the RIA algorithm.

Regarding the NLI calculation process, firstly, as in all label-free quantification experiments, we normalized the total identified DIA signals based on the sum of the heavy and light channels across time. Then, we extracted the light channel quantities (which showed a perfect pattern of gradual loss of the light intensity over time) by fitting the desired curve on each peptide. Finally, we aggregated peptide precursor k_{loss} values to the protein AS isoform groups level. For the purpose of comparison, the calculation was performed for the same data set (twelve HeLa cell lines), using the same Spectronaut results as an input for the algorithm, and the same parameters for peptide k_{loss} averaging as in the case of the RIA-based data presented in our study.

Figure 5: Correlation between RIA- and NLI-based $\log_2 k_{\text{loss}}$ and corresponding mRNA- k_{loss} values. Correlation between $\log_2 k_{\text{loss}}$ absolute values (average of all HeLa cell lines; **A**) and relative values (six HeLa Kyoto cell lines/ six HeLa CCL2 cell lines \log_2 fold change; **B**). Correlation between the across-cell lines protein AS isoform-specific mRNA- k_{loss} correlation (**C**) and corresponding mRNA- k_{loss} distributions (**D**). ρ indicates Spearman's correlation.

We first assessed whether the RIA and NLI methods provided comparable results. The \log_2 scale k_{loss} correlation analysis based on averaged value of all HeLa cells demonstrated a high correlation ($\rho = 0.95$; **Figure 5A**). The relative

correlation analysis between HeLa Kyoto and CCL2 samples also revealed that consistent results were largely achieved between RIA and NLI ($\rho = 0.77$; **Figure 5B**). This highlights the possibility of directly using light signals in a two-channel pSILAC-DIA experiment for k_{loss} calculation. Next, we calculated the proteome-wide, protein AS isoform-specific mRNA- k_{loss} correlation between all twelve HeLa cells using NLI-derived k_{loss} values. We found that despite this process involved independent calculation in each of twelve HeLa cells, the correlation coefficients still significantly correlated (**Figure 5C**), which is as strong as the relative ratio-based correlation. Finally, we assessed whether majority of the NLI-derived mRNA- k_{loss} ρ remained positive (i.e., above 0; **Figure 5D**). Indeed, although NLI generated a lower median (0.08, compared to 0.13 in RIA), 57% of the values were still above 0. This result thus supports the notion that the cells are likely to use protein degradation to fine-tune mRNA variation.

Figure 6: Comparison of RIA- and NLI-based k_{loss} precision across the HeLa cell lines. Coefficient of variation (CV) was calculated for each protein AS isoform group within six HeLa Kyoto or six HeLa CCL2 cell lines. A threshold of CV < 0.2 (red line) was highlighted.

We next asked whether NLI derives precise measurement of k_{loss} similar to RIA results. We performed a comparison of CVs of protein-level k_{loss} values generated by both methods, across replicates of CCL2 and Kyoto cells. Interestingly, we found RIA still generated more precise quantification with a statistical significance (Wilcoxon test, $p < 1 \times 10^{-16}$ for both CCL2 and Kyoto HeLa cells; **Figure 6**). This seems to suggest the calculation of k_{loss} using ratio of H and L signals (i.e. RIA) is still more stable than single L channel-based determination (i.e., NLI), possibly because the decrease of light signals is small in early time points (e.g., at 1 hour or 4.5 hours), which essentially presents a challenging case of relative quantification even for DIA-MS.

Following, we repeated the analysis presented in **Figure 3D-G** (main text in the article), but limited to k_{loss} values with a CV < 0.2 for both RIA and NLI, to achieve a fair comparison. The results were consistent to **Figure 3D-G** in the main text. As shown in **Figure 7**, the absolute mRNA- k_{loss} correlation was determined to be $\rho = -0.22$ by RIA and -0.26 by NLI (**Figure 7A&B**; $n = 2,513$; $p < 0.0001$) in all protein AS isoform groups (including both UQ and SM), and the absolute protein- k_{loss} correlation was even more negative ($\rho = -0.40$ by RIA and -0.42 by NLI; $n =$

2,513; $p < 0.0001$). Thus, the degradation seems to be slower for genes with higher expression on the absolute scale. However, when the Kyoto/CCL2 fold changes are analyzed, slight but significant positive across-gene mRNA- k_{loss} correlations were obtained, which is identical to **Figure 3** in the main text (**Figure 7C&D**; $\rho = 0.13$ for RIA, $\rho = 0.09$ for NLI; $n = 2,513$; $p < 0.0001$). Altogether, NLI and RIA were in a good agreement about absolute and relative correlations between mRNA and k_{loss} .

Figure 7: Absolute and relative correlation analysis using RIA- and NLI-based k_{loss} values. (A-B) Across-gene Spearman's absolute correlation between indicated values (average from all cell lines). **(C-D)** Spearman's correlation between indicated values using relative quantification data (Kyoto/CCL2 fold change).

Finally, we assessed the overall protein buffering significance in the NLI- and RIA-based data (Kyoto and CCL2 CV < 0.2 in both) to confirm whether the conclusions remain the same when k_{loss} is directly derived from the loss of the light, unlabeled, peptide. According to **Figure 3H** in the main text of the manuscript, we assessed the mRNA-protein and the mRNA- k_{loss} correlations for protein AS isoform groups (including both UQ and SM) that were either differentially expressed (t-test, Benjamini-Hochberg FDR < 0.01) between Kyoto and CCL2, or not (**Figure 8A**). Consistently with **Figure 3H** in the main text, we observed for both k_{loss} calculation methods proteins significantly regulated between the cell lines show a significantly lower mRNA- k_{loss} correlation than proteins which were not differentially expressed.

In **Figure 8B**, we provide mRNA- k_{loss} correlation for both methods distributed to protein complex participation (i.e., according to CORUM annotation, proteins known to be the subunits of a stable protein complex are marked as “Complex IN”, and proteins without a protein complex annotation are marked as “Complex OUT”). Because protein complex stoichiometry is a general mechanism that is well known for protein buffering, this analysis confirms mRNA- k_{loss} correlation is indeed a good measure. The better mRNA- k_{loss} correlation tend to result in more constraints at the protein level and thus worse mRNA-protein correlations (**Figure 8B**). Compellingly and conceivably, proteome wide mRNA- k_{loss} is a good indicator of protein abundance buffering mechanism independently on the method of k_{loss} calculation.

Figure 8: RIA- and NLI-based mRNA- k_{loss} protein AS isoform group-specific correlation resolved to **(A)** proteins differentially expressed between the cell line or not and **(B)** protein complex subunits and other proteins.

An additional test that should be included:

- The authors correctly claim that buffering means that for example a ten fold increase in mRNA could only lead to a four fold increase in total protein and this should rightfully be also called buffering, even so the expression level changes significantly between both conditions at mRNA and protein level.

If the authors conclusion is right that this buffering is really due to Kloss then I would like to see that proteins that show strong mRNA changes and seem buffered at the protein level indeed have a more significant kloss change than genes that have mRNA changes but the protein level changes correspond accordingly. The authors could for example bin genes based on the mRNA fold changes and then in each bin select a number of genes where the mRNA and protein changes correspond well and determine for these the Kloss to mRNA correlation and compare that to mRNA to Kloss correlation in the genes in the same bin where the mRNA changes seem buffered at the protein level.

- As a general note - any conclusions about absolute buffering have to be made cautiously as the absolute values of the relative changes matter and then the different dynamic range between RNA-seq and DIA pSILAC mass spectrometry could matter.

I have to say this is one of the cases where I wish I could talk to the authors directly to clarify. There might be a big misunderstanding on my side and I am sure that a few minutes of discussion would clarify a lot.

> Author: Firstly, we believe we already performed an analysis yielding similar conclusions as suggested by the reviewer (**Figure 3H**, and also see **Figure 8A** in **Supplementary Note 4**), and we strongly suggest that the variation of protein level measurement (**Figure 3H** and **Figure 5D** in the article) is the best, direct measure to infer the protein level buffering. In **Figures 3H** and **8A** we revealed that proteins differentially expressed between the cell lines, **a)** showed a significantly higher mRNA-protein correlation, and **b)** showed a significantly lower mRNA- k_{loss} correlation. This is essentially the same message this reviewer is seeking. Furthermore, in **Figure 5D** we show that higher mRNA- k_{loss} correlation is associated with lower protein CVs between the cell lines, independently of the corresponding protein intensities (see the new **Appendix Figure S8**), indicating the “buffering attempt” can have a global and measurable impact on protein level.

Secondly, following the reviewer request here, we performed the exact analysis (**Figure R1**). Briefly, the data were divided into five bins based on the mRNA fold change between Kyoto and CCL2. Then, two bins of genes containing the most significant fold changes (*i.e.*, the **bottom 20 %** and the **top 20 %**; together termed as “**mRNA.Sig**”) were further divided into five bins, this time based on the mRNA-protein correlation. We showed that when the **bottom 20 % (mRNA.Sig.Q1)** and the **top 20 % (mRNA.Sig.Q5)** of the mRNA.Sig genes binned based on mRNA-protein correlation are compared, both RIA- and NLI- based mRNA- k_{loss} correlations were significantly greater for **mRNA.Sig.Q1** than **mRNA.Sig.Q5**, strongly indicating that the low correlation of mRNA and protein in **mRNA.Sig.Q1** may be partially caused by a greater level of posttranslational buffering affecting the final protein levels.

Figure R1: Correlation analysis of genes with the greatest mRNA fold changes. (A) The top 20 % and bottom 20 % of genes with the most significant fold changes (**mRNA Sig**) are indicated by pink color. The “mRNA Sig” genes were further divided into five bins based on their mRNA-protein correlation, and mRNA-protein (B), RIA-based mRNA-kloss (C), and NLI-based mRNA-kloss correlations of the bottom 20 % (**mRNA.Sig.Q1**) and the top 20 % (**mRNA.Sig.Q5**) were visualized, and a statistical analysis was performed using Wilcoxon test.

A few very minor issues:

2. I don't understand the following conclusion on page 9 (last sentence is not clear):

"Correspondingly, the mRNA-protein correlation is high for differentially expressed proteins (median $\rho = 0.70$ and 0.66 , UQ and SM; Fig 3H), indicating protein level variability globally follow mRNA changes. Interestingly, only 10.96% (that is, 97 out of 885) and 12.61% (that is 365 out of 2,895) of UQ and SM proteins had a markedly regulated degradation rate (Fig 3J). To summarize, the protein degradation globally tunes protein levels but has significant buffering preferences."

> Author: The last sentence of this paragraph summarizes the main conclusions from the analysis presented in **Figure 3** in which we reported that, 1) the posttranslational buffering by k_{loss} is associated with lower protein level variation between states (**Figure 3H**), 2) only a small proportion of proteins showed differential degradation

between the two cell lines (**Figure 3**). We therefore removed this unclear sentence because the descriptions above were concise and accurate.

3. Explain the following conclusion on page 10. Is that because you see the general trend that high expressed genes have a lower k_{loss} ?

"However, the "complex I" proteins are much less abundant than "matrix" proteins. Thus, the dramatic degradation rate difference between the two may be largely ascribed to the absolute protein abundance difference (Fig EV3-B)."

> Author: The reviewer is correct. To clarify this, these sentences now read like: ***However, the "complex I" proteins are much less abundant than "matrix" proteins. Thus, the dramatic degradation rate difference between the two might be largely ascribed to the absolute protein abundance difference (Fig EV3-B) because of the general trend that highly expressed genes have a lower k_{loss} .*** (Paragraph 2 Page 9)

4. Page 11 top - please show intensity distribution of protein measurements and RNA measurements for each quintal to make sure that the difference in CV is not due to different signal to noise ratios in each quintal.

"...the top 20% proteins (Q5) with highest mRNA- k_{loss} correlation show lower coefficients of variation (CV) than the bottom 20% proteins (Q1), and both groups were significantly different from the rest of the data (Q2-Q4), confirming the modulation of protein levels by protein degradation-mediated buffering mechanisms (Kruskal-Wallis $p < 1e-10$; Fig 5D)."

> Author: Based on this reviewer's suggestion, we now add a new supplementary figure (**Appendix Figure S8**) showing intensity distributions for both mRNA and protein measurements in each mRNA- k_{loss} quintile. As indicated by the new **Figure S8**, the results and the significance of comparisons shown in **Figure 5D** are independent of corresponding mRNA and protein intensities.

The paragraph in the text now reads as: ***Intriguingly, irrespective of translational regulation and protein abundance (Appendix Figure S8), higher post-translational regulation effort (i.e., higher mRNA- k_{loss} correlation), generally succeeded in reducing the protein concentration variability between cells: the top 20% proteins (Q5) with highest mRNA- k_{loss} correlation show lower coefficients of variation (CV) than the bottom 20% proteins (Q1), and both groups were significantly different from the rest of the data (Q2-Q4), confirming the modulation of protein levels by protein degradation-mediated buffering mechanisms (Kruskal-Wallis $p < 1e-10$; Fig 5D).*** (Paragraph 2, Page 10)

5. Page 16: Small spelling error. I guess it should mean "and" and not "ad"

"Secondly, we used an mRNA abundance directed approach to identify ad quantify signals to isoform groups."

Thank you for the careful reading; the text has been corrected to: “and”.

6. Not sure what is meant by the second part of the sentence on page 16 (I guess the "whereas" confuses me a bit?)

"The transcript abundance-oriented detection of AS peptides enables the detection of exons only with appreciable read counts, whereas the transcript abundance-oriented quantification of AS enables a wider functional annotation. "

We removed “whereas”. The sentences now read like: ***The transcript abundance-oriented detection of AS peptides enables the detection of exons only with appreciable read counts (Lau et al, 2019). In addition, the transcript abundance-oriented quantification of AS enables a wider functional annotation.*** (Paragraph 1, Page 15)

References

Claydon AJ, Beynon R (2012) Proteome dynamics: revisiting turnover with a global perspective. *Mol Cell Proteomics* **11**: 1551-1565

Doherty MK, Hammond DE, Clague MJ, Gaskell SJ, Beynon RJ (2009) Turnover of the human proteome: determination of protein intracellular stability by dynamic SILAC. *J Proteome Res* **8**: 104-112

Lau E, Han Y, Williams DR, Thomas CT, Shrestha R, Wu JC, Lam MPY (2019) Splice-Junction-Based Mapping of Alternative Isoforms in the Human Proteome. *Cell reports* **29**: 3751-3765 e3755

Pratt JM, Petty J, Riba-Garcia I, Robertson DH, Gaskell SJ, Oliver SG, Beynon RJ (2002) Dynamics of protein turnover, a missing dimension in proteomics. *Mol Cell Proteomics* **1**: 579-591

Rost HL, Liu Y, D'Agostino G, Zanella M, Navarro P, Rosenberger G, Collins BC, Gillet L, Testa G, Malmstrom L, Aebersold R (2016) TRIC: an automated alignment strategy for reproducible protein quantification in targeted proteomics. *Nat Methods* **13**: 777-783

Accepted

12 February 2020

Thank you again for sending us your revised manuscript. We are now satisfied with the modifications made and I am pleased to inform you that your paper has been accepted for publication.

Corresponding Author Name: Yansheng Liu
Journal Submitted to: Molecular Systems Biology
Manuscript Number: MSB-19-9170